# The two extremes of Hansen's disease— Different manifestations of leprosy and their biological consequences in an Avar Age (late 7ᵗʰ century CE) osteoarchaeological series of the Duna-Tisza Interfluve (Kiskundorozsma– Daruhalom-dűlő II, Hungary)

Olga Spekker [1,2]*, Balázs Tihanyi[1,3], Luca Kis[1,3], Orsolya Anna Váradi[1,3], Helen D. Donoghue[4], David E. Minnikin[5†], Csaba Szalontai[6], Tivadar Vida[2], György Pálfi[1], Antónia Marcsik[1], Erika Molnár[1]

1 Department of Biological Anthropology, University of Szeged, Szeged, Hungary, 2 Institute of Archaeological Sciences, Eötvös Loránd University, Budapest, Hungary, 3 Department of Archaeogenetics, Institute of Hungarian Research, Budapest, Hungary, 4 Centre for Clinical Microbiology, University College London, London, United Kingdom, 5 Institute of Microbiology and Infection, School of Biosciences, University of Birmingham, Birmingham, United Kingdom, 6 Archaeological Heritage Protection Directorate, Hungarian National Museum, Budapest, Hungary

† Deceased.
* olga.spekker@gmail.com

## Abstract

To give an insight into the different manifestations of leprosy and their biological consequences in the Avar Age of the Hungarian Duna-Tisza Interfluve, two cases from the 7ᵗʰ-century-CE osteoarchaeological series of Kiskundorozsma–Daruhalom-dűlő II (Hungary; n = 94) were investigated. Based on the macromorphology of the bony changes indicative of Hansen's disease, **KD271** (a middle-aged male) and **KD520** (a middle-aged female) represent the two extremes of leprosy. **KD271** appears to have an advanced-stage, long-standing near-lepromatous or lepromatous form of the disease, affecting not only the rhinomaxillary region but also both upper and lower limbs. This has led to severe deformation and disfigurement of the involved anatomical areas of the skeleton, resulting in his inability to perform the basic activities of daily living, such as eating, drinking, grasping, standing or walking. The skeleton of **KD520** shows no rhinomaxillary lesions and indicates the other extreme of leprosy, a near-tuberculoid or tuberculoid form of the disease. As in **KD271**, Hansen's disease has resulted in disfigurement and disability of both of the lower limbs of **KD520**; and thus, the middle-aged female would have experienced difficulties in standing, walking, and conducting occupational physical activities. **KD271** and **KD520** are amongst the very few published cases with leprosy from the Avar Age of the Hungarian Duna-Tisza Interfluve, and the only examples with detailed macromorphological description and differential diagnoses of the observed leprous bony changes. The cases of these two severely disabled individuals, especially of **KD271** –who would have required regular and

**Data Availability Statement:** All relevant data are within the manuscript and its Supporting information files.

**Funding:** This work was supported by the University of Szeged Open Access Fund (grant number 5352) to OS. The National Research, Development and Innovation Office (Hungary) (grant number K 125561) and the "Árpád-ház Program" (grant number 39509/2018/KFSZ) of the Hungarian Ministry of Human Capacities provided funding for GP. This project received funding from the European Research Council (ERC) under the European Union's Horizon 2020 research and innovation programme (grant number 856453 ERC-2019-SyG) to TV. The funders had no role in study design, data collection and analysis, decision to publish, or preparation of the manuscript.

**Competing interests:** The authors have declared that no competing interests exist.

substantial care from others to survive–imply that in the Avar Age community of Kiskundorozsma–Daruhalom-dűlő II there was a willingness to care for people in need.

## Introduction

Leprosy, also known as Hansen's disease, is a slowly progressive, chronic, granulomatous infectious disease found mainly in humans [1–3]. It is primarily caused by an aerobic, slow-growing, rod-shaped mycobacterial species, *Mycobacterium leprae* or Hansen's bacillus [1–4]. The main causative agent of leprosy was first identified by a Norwegian physician, G.H.A. Hansen, in 1873 [1–3]; hence the terms Hansen's disease and Hansen's bacillus. In 2008, a genetically closely related bacterial species, *Mycobacterium lepromatosis* was discovered [5]. Infection with *M. lepromatosis* is quite rare, and the clinical course and forms of leprosy caused by this organism are indistinguishable from those caused by *M. leprae* [2, 4]. The portals of entry and exit, as well as the transmission pathways of leprosy bacilli are still not fully understood [1, 4]. The currently prevailing view is that the most common route of person-to-person spread of Hansen's disease is airborne transmission through upper airway secretion droplets [1–3]. The nasal mucosa is suspected of being the main portal of entry and exit for the pathogens [1]. When exposed to the leprosy bacilli, only a small proportion of individuals (5–10%) will become infected and develop clinically active leprosy [1–3]. This is because the majority of the global human population is not genetically susceptible to the disease [1, 6]. Upon becoming infected, susceptible individuals progress to asymptomatic latent infection before transitioning to symptomatic Hansen's disease [1]. Often, there is an extended incubation period, of around 5 years on average but sometimes up to 20 years, between the transmission of leprosy and the onset of its clinical signs and symptoms [1, 2]. Nevertheless, asymptomatic latent infection can spontaneously resolve [1].

Amongst individuals who progress to symptomatic Hansen's disease, the clinical presentation of the disease varies over a broad spectrum [1, 2, 7, 8]. This variability is dependent on the patient's genetically determined immune status in relation to the leprosy bacilli [2, 3, 7, 8]. At one end of the disease spectrum, there is tuberculoid leprosy that is a localised, minimally contagious, paucibacillary form of Hansen's disease with a less severe disease course [1, 2, 7–9]. In tuberculoid leprosy, patients exhibit strong cell-mediated and low humoral immune responses [2, 7–9]. At the opposite end of the disease spectrum, there is lepromatous leprosy that is a systemic, highly contagious, multibacillary form of Hansen's disease with a more severe disease course [1, 2, 7–9]. Lepromatous leprosy is characterised by a strong humoral and a low to non-existent cell-mediated immune response [2, 7–9]. Between these two immunologically stable polar types, there are the unstable borderline forms of Hansen's disease: borderline tuberculoid, borderline borderline, and borderline lepromatous [1, 2, 7, 10]. The vast majority of patients with Hansen's disease develop one of the borderline forms of the disease [2]; they have decreasing levels of cell-mediated immunity and increasing levels of humoral immunity as they move from the borderline tuberculoid to the borderline lepromatous form [7, 10]. Leprosy bacilli prefer temperatures below 37 ˚C to survive and proliferate, ideally growing in temperatures ranging from 27 ˚C to 30 ˚C [1, 2]. Therefore, Hansen's disease primarily affects the cooler, superficial areas of the human body, such as the skin, the peripheral nerves within or close to the skin, and the mucosa of the upper airways [1, 2]. Other parts of the human body, including the skeleton, can also become involved in leprosy [2, 8].

In its advanced stages, Hansen's disease can cause characteristic macromorphological bony changes, especially in the rhinomaxillary region of the face, the small bones of the hands and feet, and the long tubular bones of the arms and legs [11, 12]. These skeletal lesions enable the disease to be recognised in ancient human remains [13–15]. According to palaeopathological and palaeomicrobiological findings, leprosy is one of the oldest known infectious diseases that has been afflicting mankind for millennia [16, 17]. The oldest cases of probable Hansen's disease, based solely on macromorphological bony changes indicative of the disease, have been reported from a Late Copper Age (3780–3650 cal BCE) osteoarchaeological series from Hungary [18]. However, there are some major difficulties in diagnosing leprosy in ancient human remains [14]. In a large number of patients with Hansen's disease, the course of the disease is not sufficiently long to allow characteristic bony changes to develop before death [15, 19]. Therefore, these cases cannot be visually identified in osteoarchaeological series [15, 19]. Even if there is sufficient time for the lesions to form, the ability to detect them is to a high degree dependent on the completeness and preservation of an individual's skeletal remains [14, 15, 19, 20]. For instance, the non-recovery of the small bones of the hands and feet during excavation often hampers the recognition of leprosy cases in past human populations [19, 20]. Furthermore, relying solely on the observable macromorphological bony changes, it is very difficult to establish a definitive diagnosis of Hansen's disease [11, 12, 15, 19, 21]. This is because other pathological conditions or taphonomic processes can cause the same or similar alterations in the human skeleton, especially as bone can react in a limited number of ways [11, 12, 14, 19–21]. Nevertheless, the overall distribution of leprous bony changes, where leprosy bacilli survive and proliferate more readily in areas of the human body that are below the core body temperature, can be pathognomonic to the disease [20]. Therefore, it is the specific pattern of lesions in different areas of the skeleton rather than the individual alterations themselves that can provide a definitive diagnosis of Hansen's disease in osteoarchaeological series [12, 19]. Besides the establishment of a definitive diagnosis of leprosy, it can also be challenging to distinguishing between the different forms of the disease in the palaeopathological practice [22]. It is assumed that the type, severity, and distribution of skeletal lesions correspond to the immune status and consequently, to the form of Hansen's disease the patient had [21, 23]. In near-tuberculoid or tuberculoid leprosy associated with high resistance to leprosy bacilli, the skeleton may not be involved but if it is, leprous bony changes may occur solely in the postcranial elements [22–24]. In this case, involvement of the skeleton can either be unilateral or bilateral with a usually asymmetrical expression [22, 24]. Nonetheless, in near-lepromatous or lepromatous leprosy associated with a low resistance to the pathogens, leprous bony changes usually simultaneously affect the rhinomaxillary region of the face, the small bones of the hands and/or feet, and the long tubular bones of the arms and/or legs [22–24]. In this case, involvement of the skeleton is usually bilateral with a symmetrical expression [22, 24]. In ancient human remains, the unilateral or bilateral leprous involvement of the postcranial elements (in the form of acro-osteolysis and concentric diaphyseal atrophy of the hand and/or foot phalanges, metacarpals, and metatarsals), with no signs of rhinomaxillary bony changes, is indicative of tuberculoid leprosy [22, 24]. On the other hand, the concurrent leprous involvement of the postcranial skeleton and the rhinomaxillary region of the face is consistent with lepromatous leprosy [22, 24].

The analyses of ancient mycobacterial DNA, lipid biomarkers, and/or peptides enable the identification of leprosy bacilli in ancient human remains and thereby the confirmation of macromorphologically uncertain cases by providing an independent and clear evidence of the infection [14, 15, 21]. In addition, (sub)genotyping of *M. leprae* aDNA, reconstructed from ancient human cases with Hansen's disease, provides invaluable information about

the origins and geographical distribution of leprosy bacilli, and the migration routes of their human host over time [14, 21, 25, 26]. This is because different *M. leprae* genotypes seem to be linked to particular human populations around the world [14, 16, 27]. Recent subgenotypic data have revealed that Europe could be a key for the early spread and global dissemination of Hansen's disease [26]. Currently, two alternative scenarios explaining the high level of genetic diversity of *M. leprae* strains in mediaeval Europe exist [17, 25, 26]. 1) *M. leprae* originated in Western Eurasia, maybe even in Europe, and spread from there into the rest of the globe, with some of its strains later becoming less common or even absent from this continent but persisting in other parts of the world (e.g., East Asia and the Pacific) [17, 25, 26]. Or 2) diverse strains of *M. leprae* with different geographical origins were introduced to Europe prior and during the mediaeval period, perhaps via trade links or migrations [17, 25, 26]. It is suggested that after the fall of the Roman Empire, the successive westward migration of the nomadic Avar tribes from Central Asia or Asia Minor via the Middle East led to the separate introduction or re-transmission of different *M. leprae* strains into Eastern and Central Europe (including Hungary) during the early mediaeval period [13, 14, 16, 21, 25].

In the palaeopathological literature, only sporadic cases with bony changes indicative of leprosy [28, 29] have been published from Avar Age osteoarchaeological series from Hungary, especially from the Duna–Tisza Interfluve of the country. Some of these macromorphologically positive cases have been confirmed by palaeomicrobiological investigations. The 7th-century-CE cemetery of Kiskundorozsma–Daruhalom-dűlő II from the Duna–Tisza Interfluve is of great importance regarding the Avar Age history of Hansen's disease in Hungary (and consequently in Central Europe). This is because an exceptionally high number of cases with leprosy (**KD21**, **KD41**, **KD119**, **KD271**, **KD517**, **KD518**, and **KD520**) have been recognised in this osteoarchaeological series compared with other Avar Age cemeteries from Hungary [28, 29]. In three out of the seven cases with Hansen's disease from the 7th-century-CE cemetery of Kiskundorozsma–Daruhalom-dűlő II (**KD271**, **KD517**, and **KD518**), the aDNA analysis has confirmed the macromorphologically established diagnoses and provided invaluable information about the origins and geographical distribution of *M. leprae* in mediaeval Central Europe [14, 16, 25, 27]. On **KD517**, lipid biomarker analysis was also performed that has verified the macromorphology- and aDNA-based diagnoses [27, 30, 31]. However, the observations of the seven cases with Hansen's disease from the Kiskundorozsma–Daruhalom-dűlő II archaeological site have been only briefly summarised in several palaeomicrobiological [14, 25, 31] and palaeopathological [28] studies, and their detailed macromorphological description and differential diagnoses have not yet been provided.

To give an insight into the different manifestations of leprosy and their biological consequences in the Avar Age of the Hungarian Duna-Tisza Interfluve, two cases from the 7th-century-CE cemetery of Kiskundorozsma–Daruhalom-dűlő II (**KD271** and **KD520**) were examined, as they appear to represent the two polar types of the disease, lepromatous leprosy and tuberculoid leprosy, respectively. In the five other cases with leprosy from the Kiskundorozsma–Daruhalom-dűlő II archaeological site, the detected skeletal lesions were not sufficiently conclusive to justify their inclusion into this study, even though two of them (**KD517** and **KD518**) proved to be *M. leprae* DNA- and/or lipid-biomarker-positive [14, 16, 27, 30, 31]. The recorded bony changes indicative of Hansen's disease in **KD271** and **KD520** are presented and discussed in detail with relevance to their differential diagnoses. To reconstruct the type of leprosy and the biological consequences of the progression of the disease in the two demonstrated cases, the detected leprous bony changes are linked with palaeopathological and modern medical information.

## Materials and methods

### Materials

Between 5 June and 17 October 2003, prior to the construction works of motorway M43 that connects the city of Szeged with the Romanian border at Nagylak (Csongrád-Csanád county, southern Hungary), a salvage excavation was carried out under the direction of Tibor Paluch and Csaba Szalontai at the archaeological site of Kiskundorozsma–Daruhalom-dűlő II [32–34]. The site is geographically located north of the present-day town of Kiskundorozsma that is administratively a part of Szeged (Fig 1A) [32–34]. Besides the remains of an Árpádian Age settlement, an Avar Age cemetery with a total of 93 burials was also discovered at the eastern part of the large excavation area (32,444 m$^2$) that coincided with the trail of motorway M43 [32–35]. The fully excavated Avar Age cemetery of Kiskundorozsma–Daruhalom-dűlő II is situated at the western bank of the Maty Creek, on two low hills that run parallel to the riverbed [32–35]. Based on the associated grave goods (jewellery, belt ornaments, beads, and ceramic vessels), the Avar Age cemetery of Kiskundorozsma–Daruhalom-dűlő II was in use in the late 7$^{th}$ century CE (early/middle Avar transition period) (S1 Text) [32–34]. The 93 unearthed graves, containing the skeletal remains of 94 individuals, covered an L-shaped area that was comprised of four parallel rows of uniformly oriented, rectangular pit-graves (Fig 1B and 1C) [33, 34]. The individuals were buried in an extended supine position with the head at the northwestern end of each grave [32–34].

### Methods

The 94 human skeletons unearthed from the Avar Age cemetery of Kiskundorozsma–Daruhalom-dűlő II are currently curated at the Department of Biological Anthropology, University of Szeged (Szeged, Hungary). During the detailed anthropological investigation of the osteoarchaeological series, all skeletal remains were macroscopically examined with the naked eye.

Age at death [36–45] was estimated and sex [46] was determined applying standard macromorphological methods in biological anthropology (S1 Fig). The completeness of the 94 examined skeletons was evaluated using a three-level scale: 1) almost complete (more than 80% of the skeleton is extant); 2) relatively complete (50–80% of the skeleton is extant); or 3) very incomplete (less than 50% of the skeleton is extant). The preservation of the observable skeletal remains was also assessed on a three-level scale: 1) well-preserved (there is minimal overall *post-mortem* damage of the bone surfaces); fairly preserved (there is moderate overall *post-mortem* damage of the bone surfaces); or poorly preserved (there is significant overall *post-mortem* damage of the bone surfaces).

During the palaeopathological analysis of the 94 skeletons, the registration of bony changes indicative of leprosy was based on the following: 1) lesions in the rhinomaxillary region of the face were identified following the guidelines of Møller-Christensen and his colleagues [47], Møller-Christensen [48], and Andersen & Manchester [49]; 2) alterations of the small bones of the hands and feet were recorded considering the descriptions of Andersen & Manchester [50, 51], and Andersen and his colleagues [52, 53]; and 3) changes of the long tubular bones of the arms and legs were identified following the recommendations of Andersen and his colleagues [53], Aufderheide & Rodríguez-Martín [11], Ortner [12], and Roberts & Buikstra [54]. In our current paper, two individuals (**KD271** and **KD520**) from the Avar Age cemetery of Kiskundorozsma–Daruhalom-dűlő II were selected for discussion in detail, as their skeletal remains revealed bony changes indicative of the two extremes of Hansen's disease, lepromatous form and tuberculoid form, respectively:

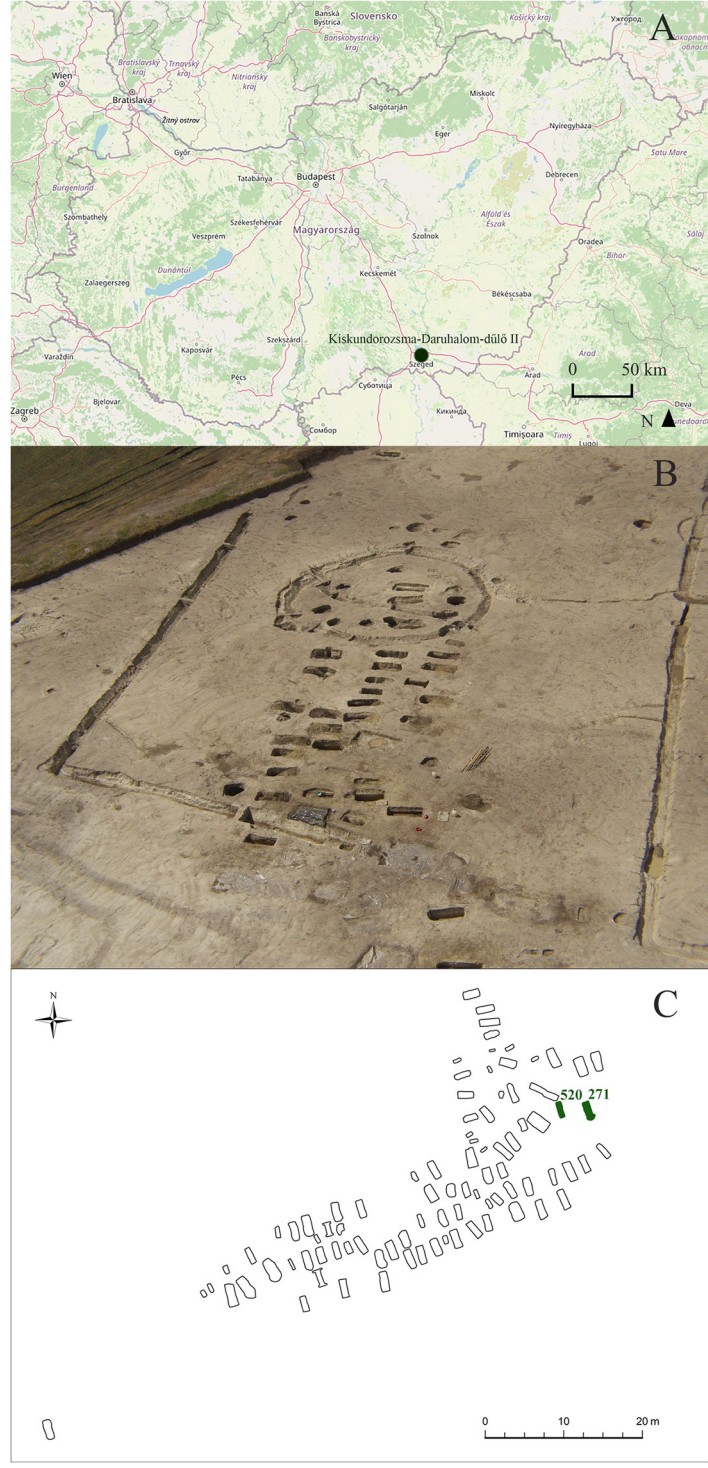

**Fig 1.** A) Map of Hungary showing the location of the Kiskundorozsma–Daruhalom-dűlő II archaeological site; B) Aerial photo of the Kiskundorozsma–Daruhalom-dűlő II archaeological site; and C) Plan drawing of the Avar Age cemetery of Kiskundorozsma–Daruhalom-dűlő II with the location of the burials of KD271 and KD520. (Contains information from OpenStreetMap and OpenStreetMap Foundation, which is made available under the Open Database License).

- **KD271 (inventory no. AP59):** a middle-aged (c. 48–57 years old) male, whose skeleton is relatively complete and well-preserved (Fig 2A and 2C); and

- **KD520 (inventory no. AP90):** a middle-aged (c. 50–59 years old) female, whose skeleton is relatively complete and well-preserved (Fig 2B and 2D).

Following the palaeopathological examination, to confirm the macromorphologically established diagnoses, aDNA analysis was also performed on the two aforementioned leprosy cases. The sample from the hard palate of **KD271** appeared to be *M. leprae* DNA-positive. Detailed information about the applied methods and results of these investigations can be found in previously published palaeomicrobiological studies [14, 25].

**Ethics statement.**   Specimen numbers: **KD271** (inventory no. AP59, grave no. 271) and **KD520** (inventory no. AP90, grave no. 520).

The two skeletons evaluated in the described study are housed in the Department of Biological Anthropology, University of Szeged, in Szeged, Hungary. Access to the specimens was granted by the Department of Biological Anthropology, University of Szeged (Közép fasor 52, H-6726 Szeged, Hungary).

No permits were required for the described study, which complied with all relevant regulations.

## Results

### KD271

In the rhinomaxillary region of the face, numerous bony changes indicative of leprosy were identified (Figs 3 and 4). The anterior nasal spine was entirely resorbed with cortical capping (Fig 4A–4D and 4F). There was bilateral, symmetrical widening and rounding of the inferior half of the pyriform aperture with resorption and remodelling of its inferior and lateral margins (Fig 4A, 4B and 4F). The inferior margins of the pyriform aperture presented vascular impressions (Fig 4D), whereas its lateral margins were smooth and thick. The maxillary alveolar process was slightly resorbed at the prosthion with involvement of the alveoli of the central incisors (Fig 4D–4F). The alveolus of the left central incisor completely disappeared and consequently, the affected tooth was lost *ante-mortem* (Fig 4E). Although it was slightly damaged *post-mortem*, the nasal surface of the maxillary palatine process exhibited extensive pitting and erosion, as well as an irregular, sharp-edged perforation on the left side (directly next to the mid-line) (Fig 4F). The oral surface of the maxillary palatine process also displayed slight abnormal pitting (maximal towards the median palatine suture) and a perforation identical with the one detected on the nasal surface (Fig 4E). Both inferior nasal conchae were entirely resorbed (Fig 4A and 4B). Although the *post-mortem* damages precluded the definitive observation of the bony nasal septum, it seems that it was at least partially (inferiorly) absorbed (Fig 4F). In the right maxillary sinus, subperiosteal new bone formations were detected on the walls of the antrum (Fig 5A). The left maxillary sinus was not observable. Finally, there were signs of caries on all extant molars and premolars, and on two extant incisors (right upper lateral and left lower central), particularly at the cervical area and the adjacent surfaces of the crowns (Fig 5B). At the cemento-enamel junction, all maxillary teeth revealed slight recession of the alveolar bone (Fig 5B).

In the postcranial skeleton, the proximal and middle phalanges of the hands (Fig 6A), the tarsal bones (Fig 6B and 6C), and the long tubular bones of the lower legs (Figs 6D, 7 and 8) exhibited bony changes that can presumably be attributed to Hansen's disease. As for the hand bones, a shallow groove was present across the entire width of the palmar surface of all extant

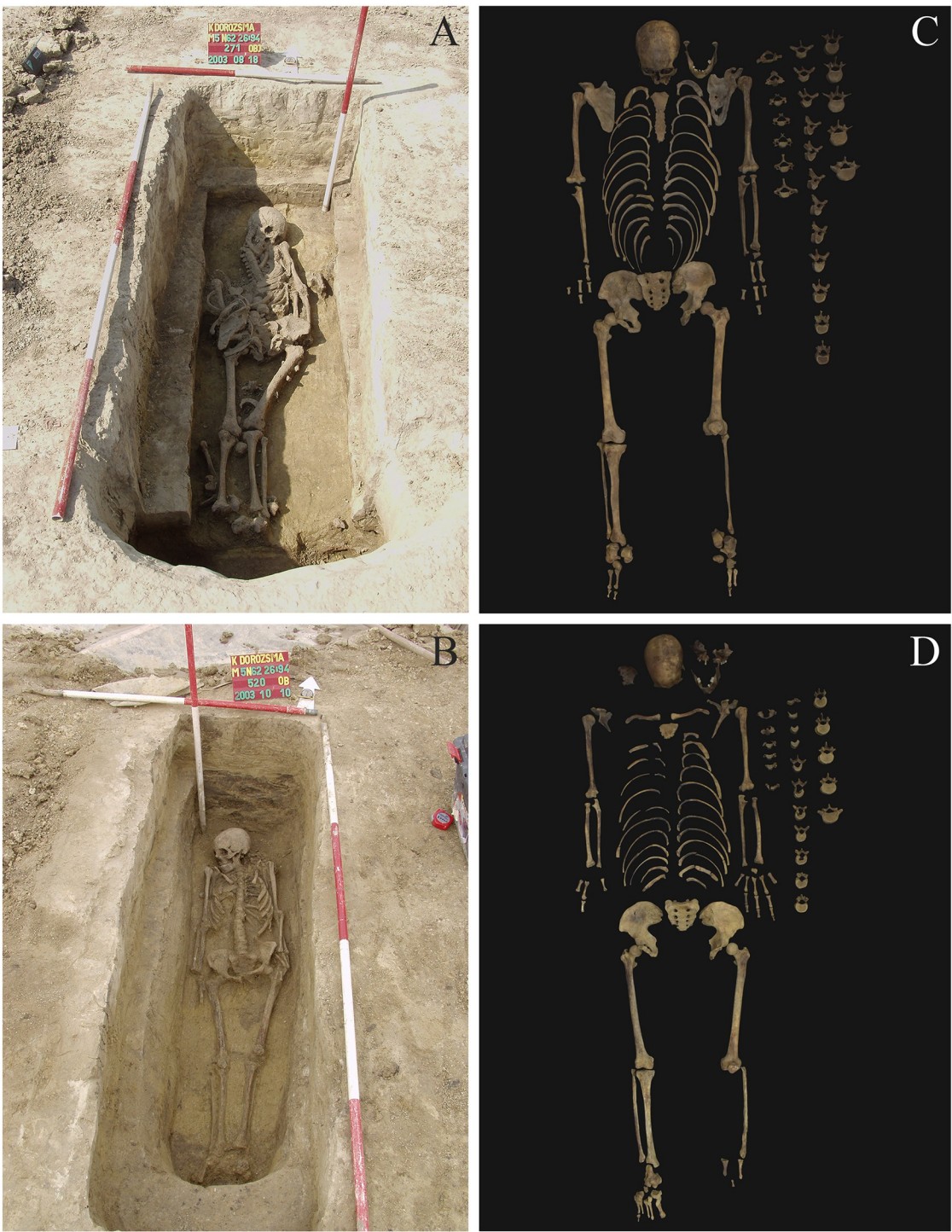

**Fig 2. Photos of the burials of A) KD271 and B) KD520** *in situ*, **and photos indicating the completeness of the skeletons of C) KD271 and D) KD520.**

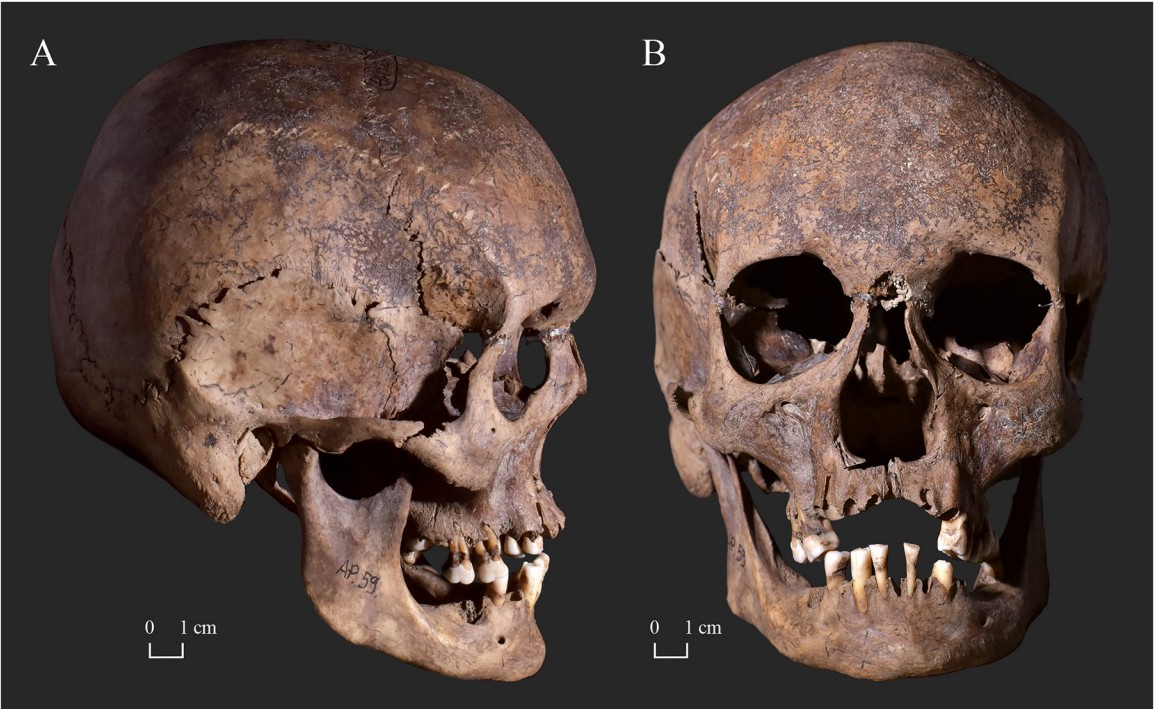

**Fig 3. A) Lateral and B) anterior view of the skull of KD271 with severe bony changes indicative of leprosy in the rhinomaxillary region of the face.**

proximal phalanges (in the juxta-articular area at the distal end) (Fig 6A). The palmar edge of the two extant middle phalangeal bases was slightly broad and flat (Fig 6A). Furthermore, marginal osteophytes were observed on the palmar surface of all extant proximal and middle phalanges, on both sides of the diaphysis (Fig 6A). In the feet, the left and right tali and navicular bones, and the left cuboid bone displayed exostoses with a height of few millimetres (at the attachment sites of the dorsal tarsal ligaments) (Fig 6B). The right cuboid bone, and the left and right cuneiform bones were missing *post-mortem*. The left and right calcanei revealed similar exostoses on their lateral surfaces (at the attachment sites of the calcaneofibular, and the lateral and interosseous talocalcaneal ligaments) (Fig 6C). Both tibiae and fibulae presented bony changes in the form of slight surface pitting, longitudinally striated subperiosteal new bone formations with a more or less organised, smooth macromorphological appearance that were seemingly continuous with the original cortical bone, and exostoses with a height of few millimetres at the attachment sites of the crural interosseous membrane (Figs 7 and 8). On both sides, the lesions were most pronounced on the adjacent surfaces of the two lower leg bones. In the tibiae, the distal half of the shaft was the most affected region (Figs 7B, 7C, 8B and 8C), whereas in the fibulae, the distal two-thirds of the shaft was the most affected (Figs 7A, 7B and 8A). Similar exostoses were detected throughout the length of the *linea aspera* of both femora, especially along the medial lip (Fig 6D).

## KD520

In the rhinomaxillary region of the face, no signs of leprosy were registered. Both orbits presented porotic *cribra orbitalia* (Fig 9A)–more pronounced on the left side. There was dental calculus, to a lesser or greater extent, on all extant maxillary (Fig 9B) and mandibular (Fig 9C)

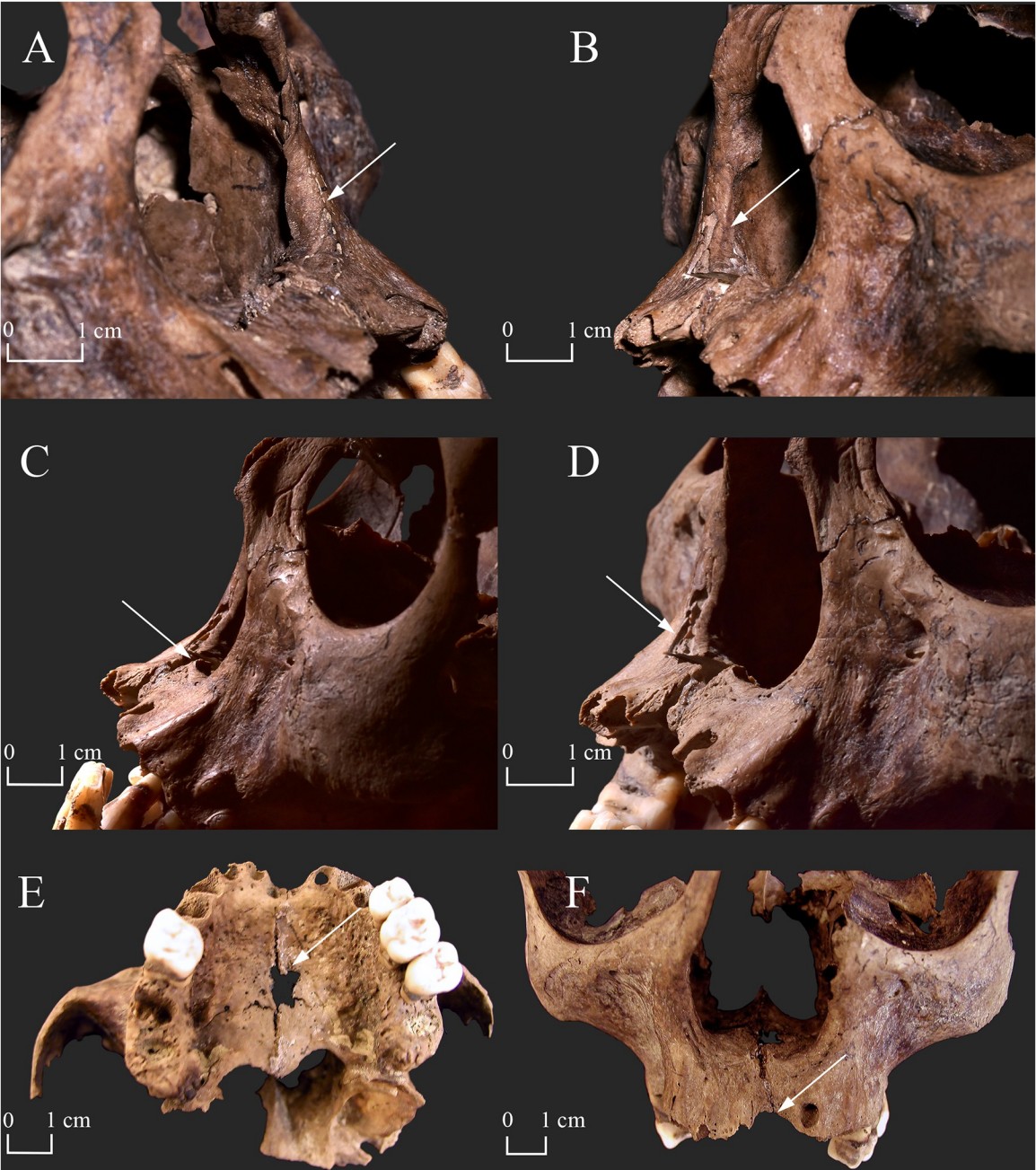

**Fig 4. Rhinomaxillary lesions in the skull of KD271.** A) Resorption of the anterior nasal spine and the left inferior nasal concha, and widening and rounding of the inferior half of the pyriform aperture (left side–white arrow); B) Resorption of the anterior nasal spine and the right inferior nasal concha, and widening and rounding of the inferior half of the pyriform aperture (right side–white arrow); C) Resorption of the anterior nasal spine (white arrow); D) Resorption of the anterior nasal spine and the maxillary alveolar process (at the prosthion), and vascular impressions in the inferior margin of the pyriform aperture (right side–white arrow); E) Pitting and perforation (white arrow) on the oral surface of the maxillary palatine process, and resorption of the maxillary alveolar process (at the prosthion); and F) Widening and rounding of the inferior half of the pyriform aperture, perforation of the nasal surface of the maxillary palatine process, and resorption of the nasal septum, the anterior nasal spine, and the maxillary alveolar process (at the prosthion–white arrow). (Photos A–D were taken after sampling for aDNA analysis, whereas photos E–F were taken before sampling for aDNA analysis).

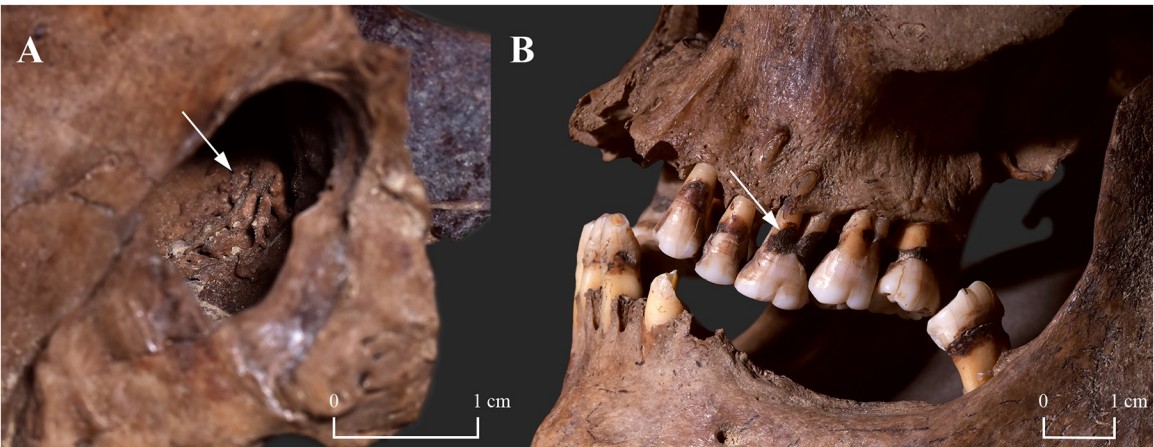

**Fig 5. A) Subperiosteal new bone formations in the right maxillary sinus (white arrow) of KD271; and B) signs of caries (white arrow) on the maxillary and mandibular teeth of KD271, and alveolar bone recession of the maxillary teeth of KD271.**

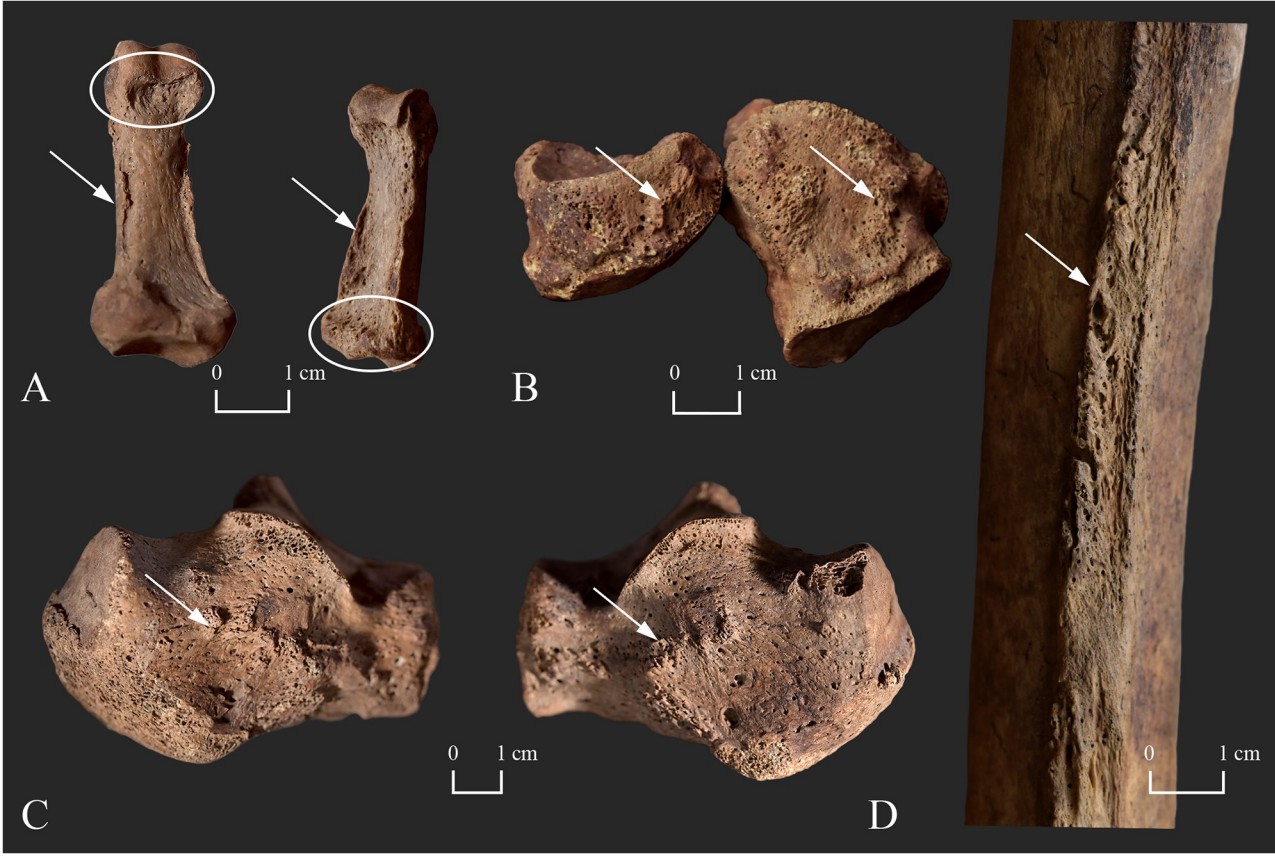

**Fig 6. Bony changes indicative of leprosy in the postcranial skeleton of KD271.** A) Shallow grooving in the juxta-articular area of the distal end of the right 5th proximal phalanx (palmar surface–white ellipse), bevelling at the base of the right 4th middle phalanx (palmar surface–white ellipse), and marginal osteophytes on both sides of the diaphysis of the right 5th proximal and right 4th middle phalanges (palmar surface–white arrows); B) Exostoses on the dorsal surface of the left navicular and cuboid bones (white arrows); C) Exostoses on the lateral surface of the right and left calcanei (white arrows); and D) Exostoses on the posterior surface of the right femur (along the *linea aspera*–white arrow).

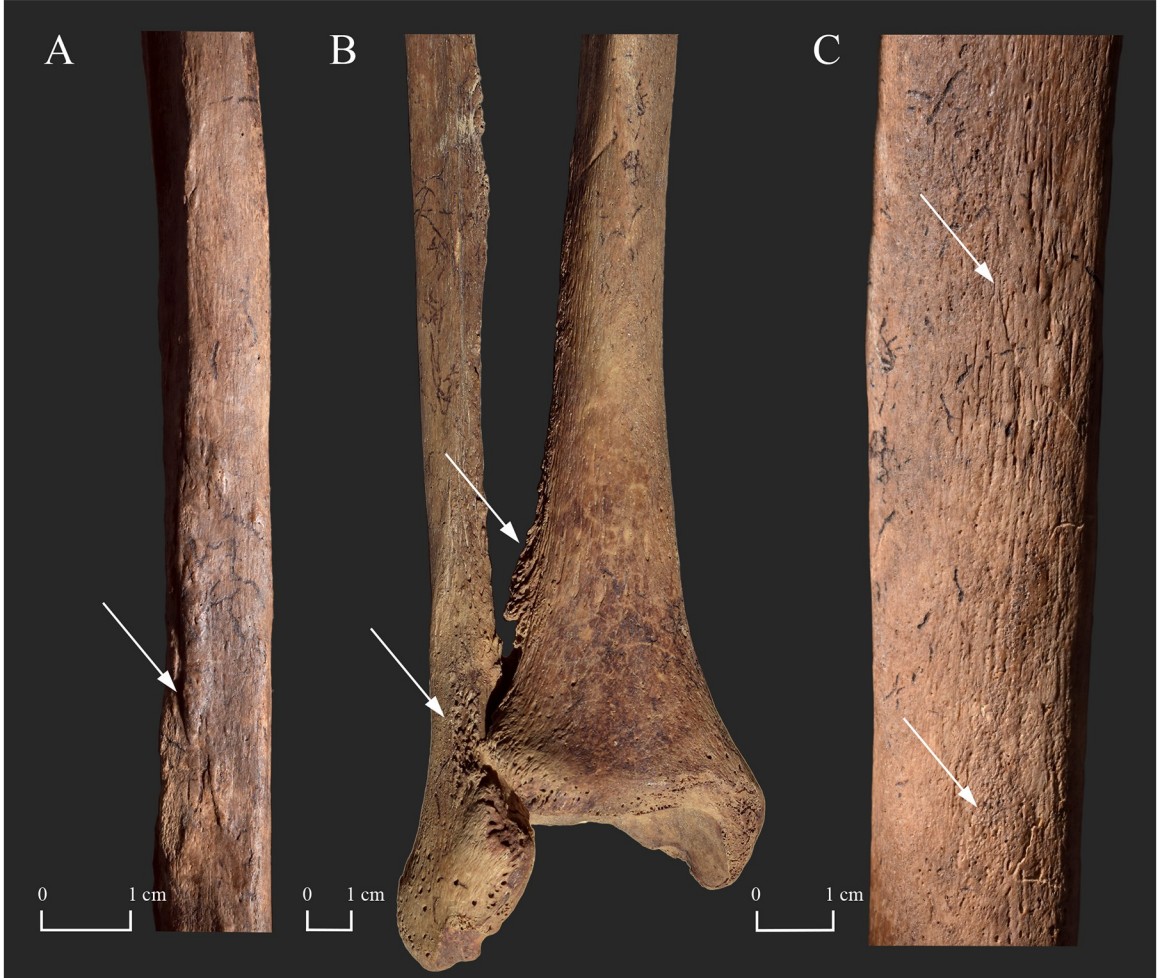

**Fig 7. A) Subperiosteal new bone formations on the shaft of the right fibula (posterior surface–white arrow) of KD271; B) Exostoses on the distal part of the right fibula and tibia (adjacent surfaces–white arrows) of KD271; and C) Surface pitting and longitudinally striated subperiosteal new bone formations on the shaft of the right tibia (medial surface–white arrows) of KD271.**

teeth. The left lower canine (Fig 9C) and the left upper 1st premolar displayed not only dental calculus but also linear enamel hypoplasia.

In the postcranial skeleton, the small bones of the feet (Fig 10) and the long tubular bones of the lower legs (Fig 11) exhibited bony changes that are suggestive of Hansen's disease. The overall diameter of the diaphysis of the right 5th proximal phalanx was decreased and its head was completely absorbed that gave the bone a pencil-like shape (Fig 10A–10C). With the exception of one undeterminable left proximal phalanx, the other foot phalanges were missing. The lateromedial diameter of the diaphysis of the right 2nd, 3rd, and 4th metatarsals was slightly decreased with the bones having a knife-shaped appearance, where the superior border of the affected diaphyses became sharp (Fig 10A, 10B and 10E). In addition, the dorsal and plantar surfaces of the proximal end of the left and right 2nd and the right 3rd metatarsals presented slight surface pitting (Fig 10E), whereas the right 4th and 5th metatarsals displayed subperiosteal new bone formations–most pronounced on the right 5th metatarsal (Fig 10C). The left 3rd, 4th, and 5th metatarsals were missing *post-mortem*. Although the substantial *post-mortem*

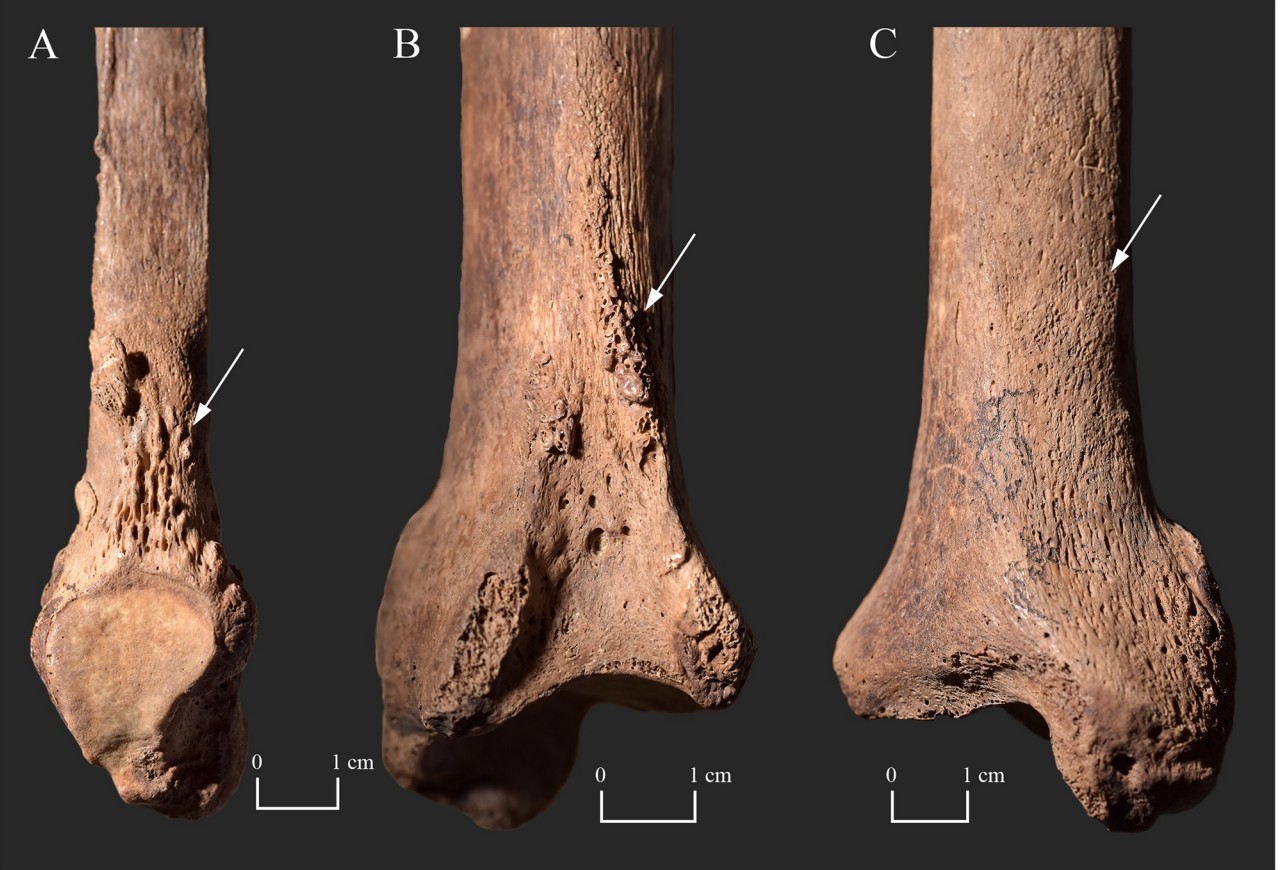

**Fig 8. Exostoses on the distal part of the right A) fibula (medial surface–white arrow) and B) tibia (lateral surface–white arrow) of KD271; and C) Slight surface pitting and subperiosteal new bone formations on the distal part of the right tibia (medial surface–white arrow) of KD271.**

damages precluded the definitive observation of the right calcaneus, its medial surface revealed pitting and subperiosteal new bone formations (Fig 10D). The left tarsal bones were missing *post-mortem*. In the lower legs, there were surface pitting and subperiosteal new bone formations throughout the length of the tibia and fibula, with the distal two-thirds of the shaft being the most affected region (Fig 11). On both sides, the lesions were most pronounced on the dorsal and adjacent surfaces of the two lower leg bones (Fig 11). The macromorphological appearance of the tibial and fibular subperiosteal new bone formations was predominantly organised and smooth, with some patches of less organised, porous new bone.

## Discussion

Although involvement of the skeleton can occur at and between both ends of the disease spectrum of Hansen's disease, the more extensive and severe bony changes can be observed in cases with lepromatous leprosy [12, 20, 54–56]. Leprous skeletal lesions can be divided into three main groups: 1) specific bony changes that develop following direct invasion of the affected bone(s) by leprosy bacilli; 2) non-specific bony changes that are secondary to leprous sensory, motor or autonomic peripheral neuropathy; and 3) osteoporotic bony changes that result from disuse [11, 20, 56].

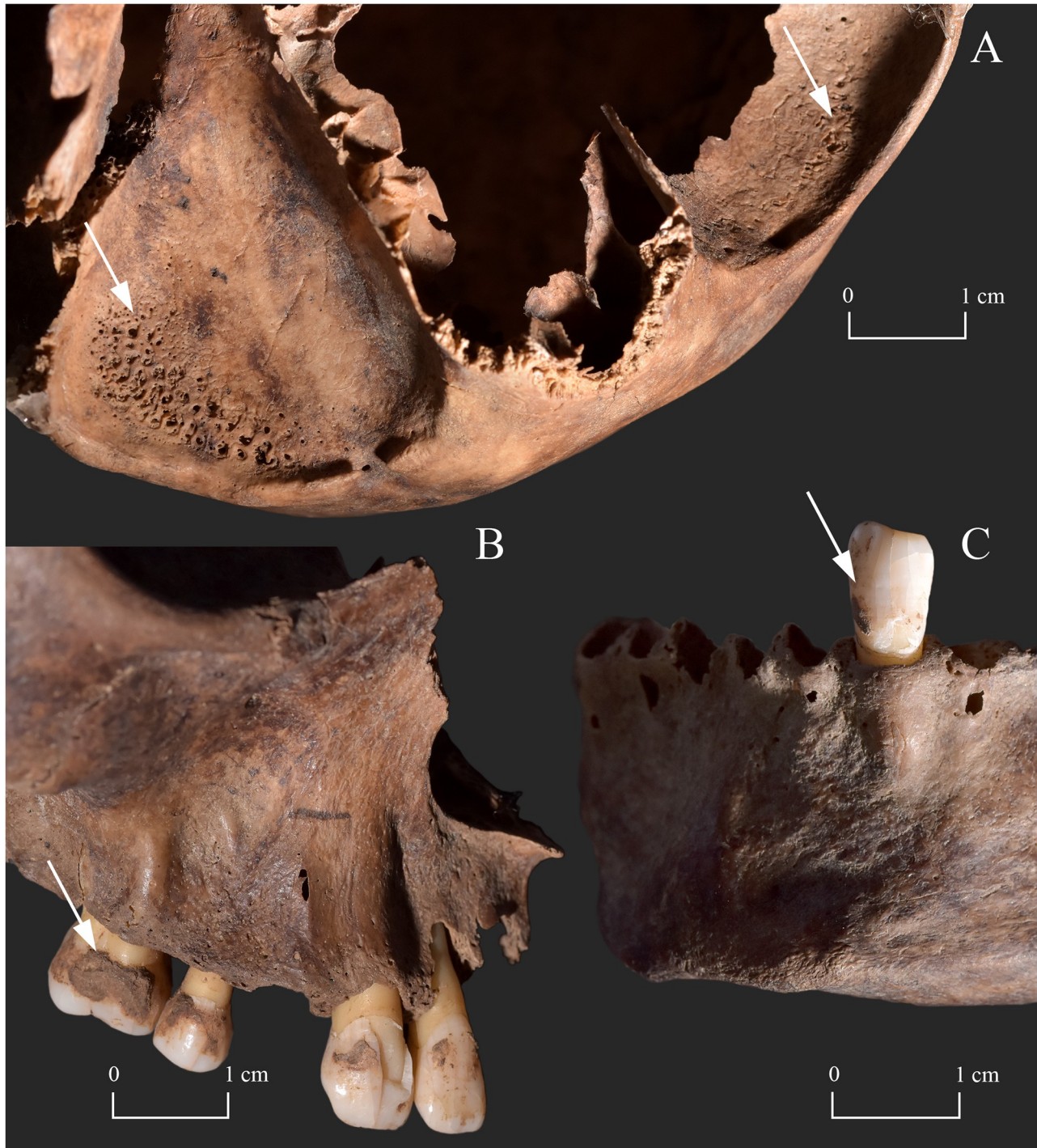

**Fig 9. A) Porotic** *cirbra orbitalia* **in the left and right orbits of KD520 (white arrows); B) Dental calculus on the maxillary teeth of KD520 (white arrow); and C) Dental calculus and linear enamel hypoplasia (white arrow) on the left lower canine of KD520.**

## Bony changes of the skull

In the skull, leprosy induces the formation of bony changes that are confined to the rhinomaxillary region of the face and are collectively referred to as '*facies leprosa*' [47] or 'rhinomaxillary

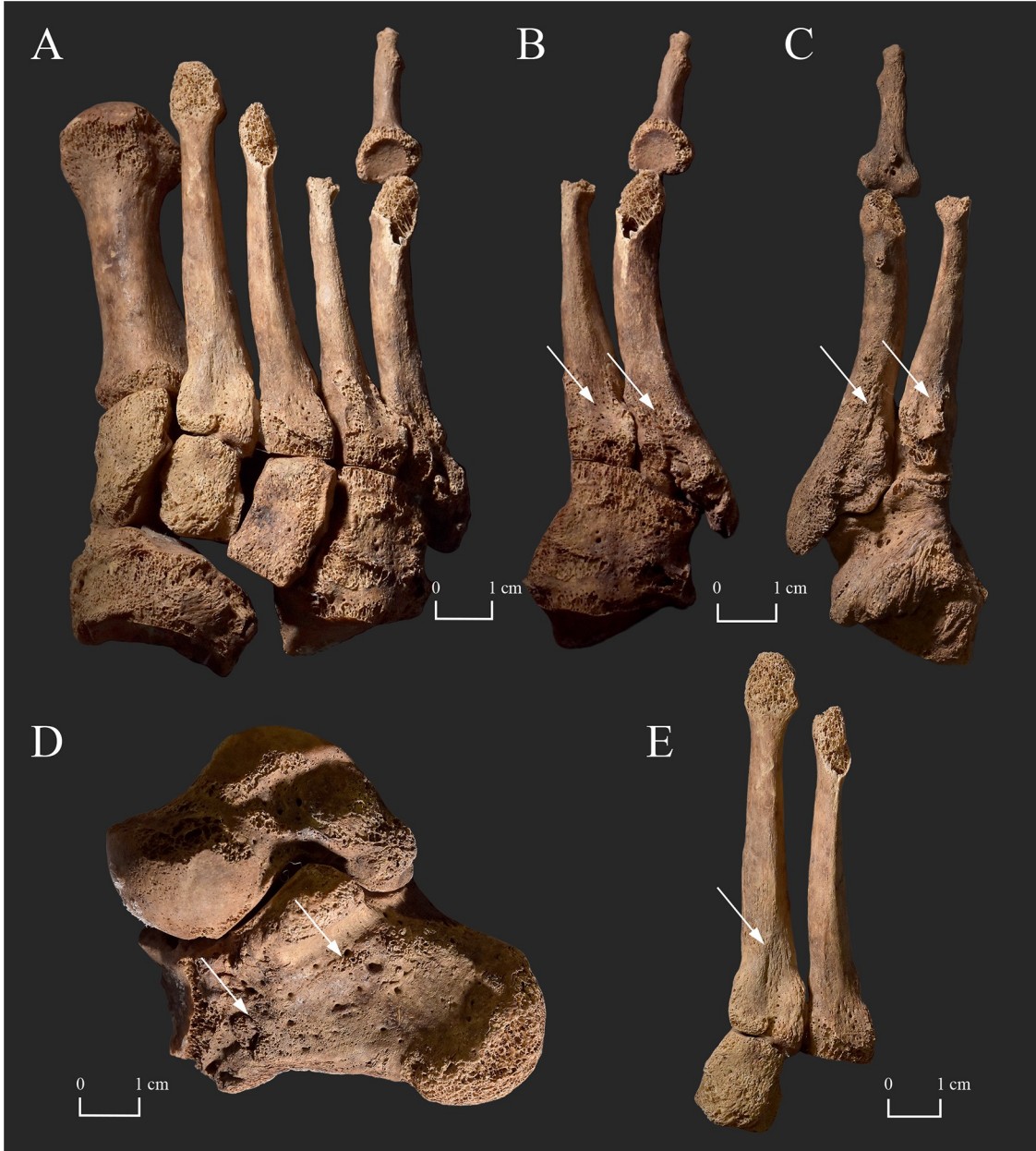

**Fig 10. Bony changes indicative of leprosy in the tarsal bones and metatarsals of KD520.** A) Concentric diaphyseal atrophy of the right 2nd, 3rd, and 4th metatarsals (knife-edge type), and the right 5th proximal phalanx (true pan-circumferential concentric type); B) Concentric diaphyseal atrophy of the right 4th metatarsal (knife-edge type) and the right 5th proximal phalanx (true pan-circumferential concentric type), and subperiosteal new bone formations on the proximal end of the right 4th and 5th metatarsals (dorsal surface–white arrows); C) Concentric diaphyseal atrophy of the right 5th proximal phalanx (true pan-circumferential concentric type), and subperiosteal new bone formations on the proximal end of the right 4th and 5th metatarsals (plantar surface–white arrows); D) Pitting and subperiosteal new bone formations on the medial surface of the right calcaneus (white arrows); and E) Slight surface pitting on the proximal end of the right 2nd and 3rd metatarsals (dorsal surface–white arrow).

syndrome' [49]. This set of lesions (S2 Text) is caused by local invasion of the affected bone(s) by leprosy bacilli [11, 20]. It can happen through direct extension of the infection from the adjacent soft tissues, such as the overlying skin and oronasal mucosa, or via haematogenous spread of the pathogens [11, 20, 55]. The rhinomaxillary syndrome is a composite mixture of

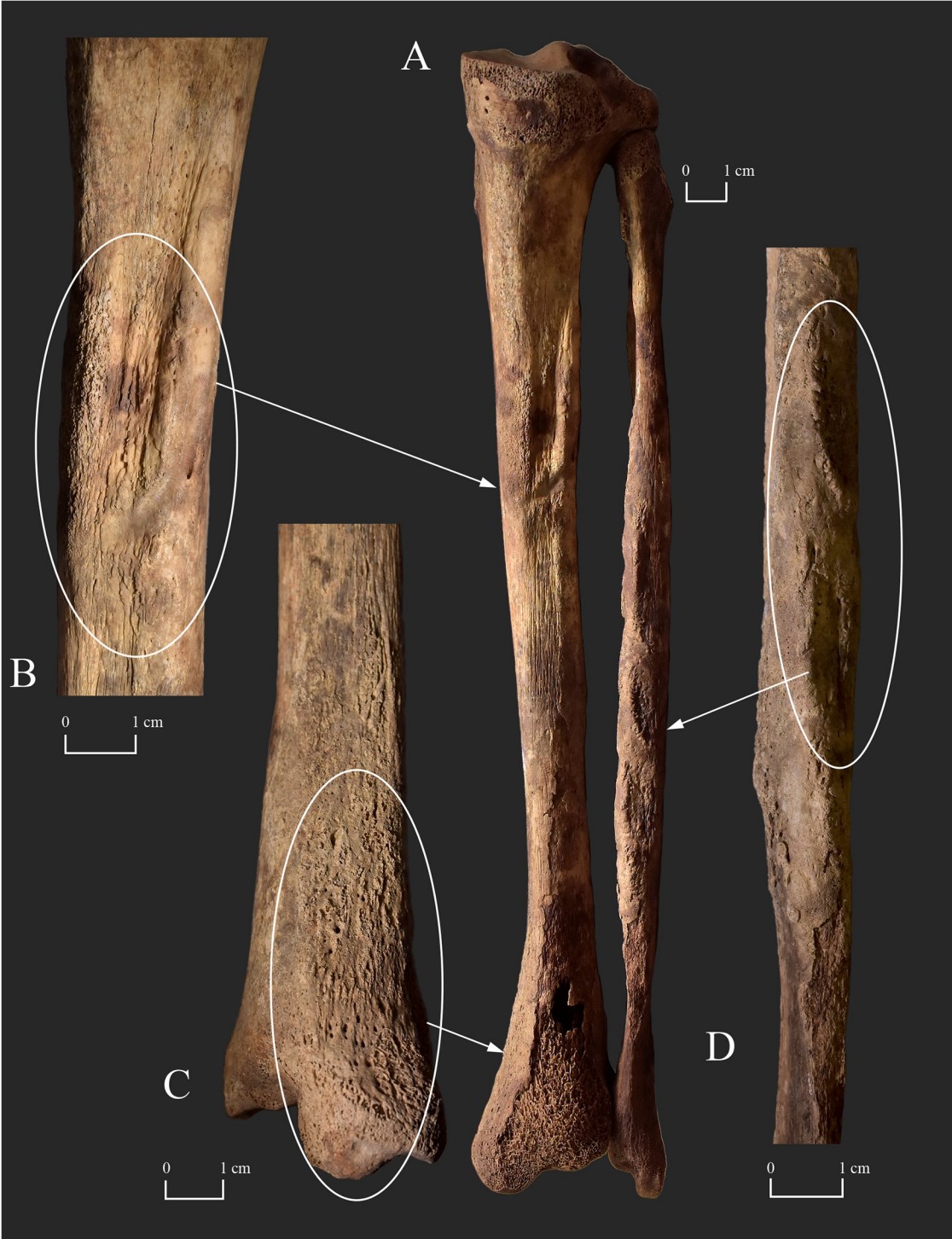

**Fig 11. Surface pitting and subperiosteal new bone formations on the right tibia and fibula (white ellipses) of KD520.** A) Dorsal view of the right tibia and fibula, and close-up of the B) shaft of the right tibia (posterior surface), C) distal part of the right tibia (medial surface), and D) shaft of the right fibula (posterior surface).

absorptive, erosive, and proliferative bony changes (S2 Text) [49]. The distribution of these skeletal lesions reflects the cooler temperature of exposed skin and mucosa that is consistent with the well-known temperature preference of leprosy bacilli [11, 20, 49, 54]. From the initial site of involvement (the nasal cavity), the leprous infection can extend to the paranasal sinuses, particularly to the maxillary ones [57]. Although not pathognomonic for Hansen's disease, signs of sinusitis in the form of pitting and/or subperiosteal new bone formations can frequently be observed on the walls of the affected maxillary sinus(es) in cases with rhinomaxillary syndrome [20, 54, 58, 59].

From the two presented cases from the Avar Age osteoarchaeological series of Kiskundorozsma–Daruhalom-dűlő II, only **KD271** exhibited bony changes in the rhinomaxillary region. To a lesser or greater extent, all the aforementioned anatomical components of the rhinomaxillary syndrome (S2 Text) were present in the middle-aged male's skull (Figs 3 and 4). This lesion combination is pathognomonic for near-lepromatous or lepromatous leprosy [11, 19, 49, 53]. In **KD271**, the initial site of leprous involvement could have been the nasal mucosa. Its infiltration by leprosy bacilli could have resulted in the onset of granulomatous inflammation with subsequent development of atrophic rhinitis (progressive thinning and hardening of the mucosa that lines the nasal cavity) [11, 49, 57, 60, 61]. This condition is characterised by highly bacilliferous, muco-purulent or exudative nasal discharges, crusting, intermittent epistaxis, and obstruction of the nasal airways [49, 57, 60, 61]. In the middle-aged male, the infection could have extended from the nasal mucosa to the adjacent cartilaginous and bony nasal structures (e.g., inferior nasal conchae and nasal surface of the maxillary palatine process), and paranasal sinuses (e.g., right maxillary sinus) [49, 57, 60, 61]. In **KD271**, destruction of the affected cartilaginous and bony nasal structures could have led to the development of saddle-nose deformity that is characteristic of lepromatous leprosy [1, 11, 49, 62]. Saddle-nose deformity can be associated with functional difficulties (e.g., impaired nasal breathing) and facial disfigurement [49, 62]. Persistent blockage of the nasal airways could have made it difficult for the middle-aged male to breathe through his nose. Consequently, he could have developed an oral breathing habit that is commonly observed in patients with lepromatous leprosy today [63, 64]. Breathing through the mouth, due to the cooling effect of the inspired air and evaporation, results in a decrease in the surface temperature of certain areas of the oral cavity (e.g., soft and hard palates, and anterior part of the maxilla) [63–66]. The lower surface temperature makes the oral mucosa of these sites more susceptible to seeding by leprosy bacilli and consequently, to developing oral leprous lesions with or without involvement of the underlying bone (e.g., premaxilla and oral surface of the maxillary palatine process) [63, 64, 66]. Besides contiguous spread, haematogenous dissemination of the pathogens from the affected nasal cavity may also be responsible for the formation of oral leprous lesions [55, 65]; oral alterations, such as perforation of the hard palate, are usually late phenomena of lepromatous leprosy [49, 63, 64]. In **KD271**, leprous involvement of both the nasal and oral surfaces of the maxillary palatine process with ultimate perforation of the hard palate could have resulted in the formation of an oronasal fistula (an abnormal communication between the oral and nasal cavities) [67, 68]. Its presence could have led to functional difficulties, such as nasal regurgitation of food or liquids and speech impairments [64, 67, 68]. In summary, the severity and extent of the observed rhinomaxillary bony changes indicate that the middle-aged male suffered from an advanced-stage, long-standing near-lepromatous or lepromatous leprosy with serious biological consequences.

Although, based on the modern medical and palaeopathological literature, Hansen's disease seems to be the most likely underlying cause of the skeletal lesions detected in the rhinomaxillary region of the face of **KD271**, other aetiologies should also be considered in the differential diagnosis. The most relevant ones are bacterial granulomatous infections other than leprosy

(e.g., treponematoses, tuberculosis, and actinomycosis), systemic fungal infections (e.g., aspergillosis and mucormycosis), and sarcoidosis (S3 Text) [11, 12, 21, 54].

In a few ancient leprosy cases with severe rhinomaxillary lesions [69–72], a short root anomaly of the permanent upper incisors, leprogenic odontodysplasia or 'dens leprosus', was concomitantly observed. Leprogenic odontodysplasia probably results from direct invasion of the dental pulp cavity by leprosy bacilli, and is characterised by the presence of a concentric constriction groove in the root and abrupt tapering with arrested tooth growth [56, 70, 72]. Leprogenic odontodysplasia can develop when the onset of leprous skeletal involvement coincides with the root formation of the affected tooth that indicates childhood exposure to the pathogens [56, 70, 72].

In the two demonstrated cases from the 7th-century-CE cemetery of Kiskundorozsma–Daruhalom-dűlő II, no signs of leprogenic odontodysplasia were detected. Nonetheless, both **KD271** and **KD520** presented other tooth pathologies (dental caries and signs of periodontitis or dental calculus, respectively) that are not specific to Hansen's disease but can frequently be observed in leprosy patients today [68, 73]. Oral breathing can play a role in increasing the risk of dental erosion, calculus, and caries formation, and halitosis in patients with Hansen's disease, as besides lowering the surface temperature of certain areas of the oral cavity, it can also influence changes in the intraoral humidification, pH, and oxygen levels [74]. Therefore, leprosy cannot be excluded from being at least partly responsible for the development of dental caries and periodontitis in **KD271**, where oral breathing can be presumed based on the recorded rhinomaxillary lesions.

## Bony changes of the postcranial skeleton

In the postcranial skeleton, most leprous lesions are not secondary to direct invasion of the affected bone(s) by leprosy bacilli but to neuropathy in the motor, sensory, and/or autonomic peripheral nerves [11]. The loss of nerve function is a consequence of inflammatory changes subsequent to the invasion of the peripheral nerve tissue by the pathogens [11, 50]. The large, superficial peripheral nerves (e.g., ulnar, median, lateral popliteal, and posterior tibial) are characteristically affected in Hansen's disease [20, 51]. Based on the above, bony changes of the small bones of the hands and feet, and the long tubular bones of the arms and legs indicative of leprosy can be considered as highly characteristic but not pathognomonic features of the disease [53].

**Leprosy-specific bony changes.** Leprous septic osteitis and osteomyelitis of the postcranial bones are rare phenomena that tend to occur in the small bones of the hands and feet [11, 12]. Both conditions are caused by direct invasion of the affected bone(s) by leprosy bacilli, usually through direct extension of the infection from adjacent soft tissues or less frequently via haematogenous dissemination of the pathogens [11, 12, 53]. Lodgement of leprosy bacilli in the bone leads to the onset of its granulomatous inflammation [53]. Leprous septic osteitis and osteomyelitis usually occur as lytic or cystic lesions in the affected cortical and/or cancellous bone [53]. These alterations are particularly located in the subchondral region of the bone underlying the articular cartilage; occasionally ballooning of the affected bone(s) with or without cortical thinning can also ensue [12, 53]. Intra-articular extension of a lytic or cystic leprous skeletal lesion into the adjacent joint gives rise to leprous septic arthritis [11]. Much less frequently, leprous septic arthritis results from haematogenous spread of leprosy bacilli into the synovial membrane of the affected joint [11].

In the two presented cases (**KD271** and **KD520**) from the Avar Age osteoarchaeological series of Kiskundorozsma–Daruhalom-dűlő II, no signs of leprous septic osteitis, osteomyelitis or arthritis of the postcranial bones or joints were recorded.

**Non-specific bony changes indicative of motor peripheral neuropathy.** Leprous dysfunction of the motor peripheral nerves innervating the hands and feet, that occurs in more advanced stages of Hansen's disease, leads to progressive paralysis of individual muscles and muscle groups with secondary deformation and disfigurement of the affected limb(s) [50, 54, 75–78]. For instance, loss of motor function in the ulnar and posterior tibial nerves can result in the development of 'claw-hand' or 'flat-foot' deformity, respectively. Detailed information regarding the bony changes that are indicative of the aforementioned deformities and consequently, motor peripheral neuropathy can be found in S4 Text.

From the two demonstrated cases from the 7th-century-CE cemetery of Kiskundorozsma–Daruhalom-dűlő II, **KD271** revealed signs of motor peripheral neuropathy both in the hand and foot bones (Fig 6A and 6B). The shallow palmar grooving at the distal end of all the extant left and right proximal phalanges and the slight broadening and flattening of the palmar edge of all the observable left and right middle phalangeal bases are consistent with long-standing claw-hand deformity of both hands [50]. It should be noted that claw-hand deformity is associated with only soft tissue contracture in its early stages [20, 50]. In the middle-aged male, the shallow grooves were present across the entire width of the palmar surface of the affected left and right proximal phalanges, implying that there was no lateral deviation in his involved proximal interphalangeal joints [50]. Claw-hand deformity, secondary to motor impairment in the ulnar nerve, is usually more evident in the 4th and 5th fingers (partial claw-hand) but the 1st, 2nd, and 3rd fingers can also become clawed (complete claw-hand) [77, 78]. The presence of marginal osteophytes on both sides of the proximal and middle phalangeal diaphyses of **KD271** could refer to permanent contracture of the soft tissues in both of his hands, since these alterations were described as entheseal changes of the superficial finger flexor muscles [79, 80]. In the middle-aged male, deformation could have led to disfigurement of the hands and deterioration of the hand functions (e.g., weakened grasp, grip, and pinch) [81]. Thus, he would have experienced disability in performing the basic activities of daily living [78, 82]. It is important to bear in mind that claw-hand deformity can either be congenital or acquired–it can be caused by injuries or disorders that lead to neuropathy of the ulnar nerve (e.g., Charcot-Marie-Tooth disease, burn injury to the hand or Hansen's disease) [78, 83, 84]. Therefore, the presence of claw-hand deformity cannot be considered as a pathognomonic feature of leprosy [54]. Nevertheless, palmar grooving of the distal end of the proximal phalanges and coincident palmar bevelling of the corresponding middle phalangeal bases are not known to be associated with claw-hand deformity of other aetiologies in the literature [50]. Furthermore, the co-occurrence of the aforementioned postcranial lesions with rhinomaxillary bony changes indicative of Hansen's disease also supports their leprous origin in **KD271**.

In the left and right tarsi of **KD271**, the presence of exostoses at the attachment sites of the dorsal talonavicular, cuneonavicular, and cuboideonavicular ligaments is consistent with flat-foot deformity of both feet secondary to leprous involvement and consequent impairment of the motor component of the posterior tibial nerve [51]. With the exception of the left 2nd and the right 2nd and 3rd proximal phalanges, the foot phalanges are missing; thus, precluding their evaluation regarding the presence of claw-toe deformity that can also ensue subsequent to motor neuropathy in the posterior tibial nerve. In the middle-aged male, the bilateral development of flat-foot deformity could have resulted in foot disfigurement and difficulties in standing and walking, and thereby in conducting domestic and occupational physical activities [85]. Flat-foot deformity can either be congenital or acquired–besides leprosy, injuries (e.g., rupture of the posterior tibial tendon) or disorders other than Hansen's disease (e.g., congenital tarsal coalition or *diabetes mellitus*) can also give rise to its development [51, 54, 86–88]. Nonetheless, the formation of dorsal tarsal exostoses has not been considered to be a sequel of flat-foot deformity of other aetiologies in the literature [51]. The dynamic, progressive, plantar

displacement of the navicular bone rather than a static state of the altered anatomical architecture of the feet may stimulate the development of these exostoses in Hansen's disease [51]. In addition, the presence of dorsal tarsal exostoses with other postcranial lesions suggestive of leprous peripheral neuropathy and with rhinomaxillary bony changes indicative of Hansen's disease supports their relationship with the disease in **KD271**. Bilateral, symmetrical, distal polyneuropathy of the peripheral nerves, that can be presumed in the case of the middle-aged male from the Kiskundorozsma–Daruhalom-dűlő II osteoarchaeological series, is characteristic of lepromatous leprosy [1, 89].

**Non-specific bony changes indicative of sensory peripheral neuropathy.** Leprous dysfunction of the sensory peripheral nerves innervating the hands and feet leads to diminution or complete loss of one or more sensation modalities [50, 53, 90, 91]. Early in the course of leprosy, cutaneous sensory impairment occurs that predisposes patients to unintentional minor superficial trauma and secondary pyogenic sepsis of the involved skin area(s) [11, 23, 53, 90]. In more advanced stages of Hansen's disease, not only the cutaneous sensation but also the deep tissue sensation will be lost, and the pyogenic sepsis extends from the superficial tissues to the deep soft tissues, bones, and joint cavities [11, 23, 50, 53]. This gives rise to pyogenic septic periostitis, osteitis, osteomyelitis, and/or arthritis of the small bones and joints of the hands and feet with consequent formation of bony changes [11, 50, 53]. Detailed information regarding the aforementioned skeletal lesions can be found in S5 Text.

In **KD271**, the skeletal lesions of the left and right tibiae and fibulae (surface pitting and longitudinally striated subperiosteal new bone formations) imply that not only the motor but also the sensory component of the posterior tibial nerve, that supplies sensation to the plantar foot, was affected by Hansen's disease in both lower limbs (Figs 7 and 8). After the loss of cutaneous sensation, plantar ulcer(s) could have developed in the middle-aged male's feet [92]. Even the customary use of the insensitive feet in standing and walking could have caused their plantar ulceration, particularly in the weight-bearing areas [20, 53]. If there is dysfunction of the motor component of the posterior tibial nerve with consequent collapse of the longitudinal arch and formation of flat-foot deformity, as it can be presumed in the case of **KD271**, plantar ulcers tend to occur in the mid-foot (in correlation with the altered plantar weight distribution of the body weight consequent to foot deformation) [53]. It is important to bear in mind that signs of periostitis in the lower leg bones appear to develop more frequently when accompanied by mid-foot plantar ulceration [20]. Although some of the foot bones of **KD271** are absent and the extant ones do not display pyogenic septic bony changes, which would directly indicate the loss of deep tissue sensation and the *in vivo* presence of plantar ulcer(s), it is permissible to infer their presence in both feet of the middle-aged male: 1) impairment of the motor component of the posterior tibial nerve, that can be presumed in both of the middle-aged male's feet, is generally accompanied by regional sensory loss [53, 76]; and 2) in most cases with leprosy, skeletal lesions suggestive of tibial and/or fibular periostitis do not occur in the absence of plantar ulceration [23, 53]. In **KD271**, the inflammatory bony changes observed in the lower leg bones could have resulted from a pyogenic infection ascending from one or more plantar (mid-foot) ulcers before the pathological process could have involved the foot bones underlying the affected soft tissues [53]. The more organised, smooth, lamellar-like macromorphological appearance of the observed subperiosteal new bone formations is indicative of an infection that occurred and healed before the middle-aged male's death as opposed to a less organised, porous, woven-like one that would represent an active phase of the infection at the time of death [23, 93]. The bilateral presence of exostoses at the attachment sites of the crural interosseous membrane throughout the length of the tibial and fibular shafts indicate that this ligamentous structure was also inflamed on both sides due to the pyogenic infection [23]. Although the hand bones of **KD271** do not provide any skeletal evidence for the

presence of sensory neuropathy in the ulnar nerve, motor dysfunction of this nerve, that can be presumed in both of the middle-aged male's hands, can indirectly indicate it. This is because in leprosy, motor impairment is generally preceded by cutaneous sensory loss and accompanied by regional sensory loss [53, 76].

Similar to **KD271**, the inflammatory skeletal lesions (surface pitting and subperiosteal new bone formations) observed in the tarsal bones and metatarsals (Fig 10), as well as in the tibiae and fibulae (Fig 11) of **KD520** imply bilateral leprous involvement and consequent impairment of the sensory component of the posterior tibial nerve. The pyogenic septic bony changes in the middle-aged female's tarsal bones and metatarsals, especially of the right calcaneus and the right 5th metatarsal, serve as direct evidence for the *in vivo* presence of plantar ulceration of both of her feet [53]. Based on the presence of alterations not only on the plantar but also on the dorsal surface of the extant metatarsals, both foot surfaces could have concomitantly been affected by ulcers. Furthermore, the presence of the above-mentioned skeletal lesions indicates that, in both lower limbs of **KD520**, not only the cutaneous but also the deep tissue sensation could have been lost in the feet consequent to sensory dysfunction in the posterior tibial nerve. This is because the pyogenic sepsis extends from the superficial tissues to the bones following the development of deep tissue anaesthesia [53]. In contrast to **KD271**, no dorsal tarsal exostoses indicating longitudinal arch collapse and foot deformation were observed in the extant tarsal bones of the middle-aged female's right foot [53]. The localisation and severity pattern of the pyogenic septic bony changes in the right tarsal bones and metatarsals of **KD520** suggest that the *in vivo* plantar ulcers could have particularly been present at the sole and the 5th tarsometatarsal joint–in close vicinity to two of the three main weight-bearing areas of the foot in normal plantar weight distribution of the body weight [53, 94]. This suggests that the normal anatomical architecture of the insensitive right foot could have been intact or the collapse of its longitudinal arch could have been quite recent when the middle-aged female died [53]. Based on the above, even if there was a loss of motor function in the right posterior tibial nerve, it seems that there was no time for it to evidently manifest itself in the skeleton before the death of **KD520**. It should be noted that the same conclusions cannot be drawn for the left lower limb as its tarsal bones and the majority of its metatarsals are missing *post-mortem*. Similar to **KD271**, the inflammatory bony changes observed in the middle-aged female's tibiae and fibulae could have resulted from a pyogenic infection ascending from the ulcers of her feet [53]. The mixed macromorphological appearance of the subperiosteal new bone formations–predominantly well-organised, smooth, and lamellar-like new bone accompanied by small patches of disorganised, porous, and woven-like new bone–is considered indicative of a chronic infection that was active at the time of death [23].

It is important to note that apart from Hansen's disease, a number of medical conditions (e.g., treponematoses and tuberculosis) can result in periostitis and subsequent development of bony changes in the small bones of the hands and feet and/or the long tubular bones of the lower legs [19, 23, 54, 93]. Even in leprosy patients, these inflammatory alterations can result from not only a secondary pyogenic infection but other aetiologies, such as trauma to the periosteum or direct spread of *M. leprae* to the periosteum from a leprous soft tissue lesion overlying the bone [20, 23]. If the distal end of the lower leg bones rather than their mid-shaft or proximal end is the most severely affected region by the skeletal lesions indicative of periostitis, the diagnosis of Hansen's disease should be considered [95]. Moreover, the presence of the inflammatory bony changes observed in the long tubular bones of the lower legs and/or the small bones of the hands and/or feet of **KD520** and **KD271** with other alterations of which the leprous origin is much more certain (concentric diaphyseal atrophy in **KD520** and rhinomaxillary lesions in **KD271**) supports their relationship with Hansen's disease in the middle-aged female and male from the 7th-century-CE cemetery of Kiskundorozsma–Daruhalom-dűlő II.

**Non-specific bony changes indicative of autonomic peripheral neuropathy.** Leprous dysfunction of the autonomic peripheral nerves innervating the hands and feet is accompanied by circulatory disturbances and consequent changes in the blood oxygen tension of the affected area(s) [12, 52, 54]. This, by selectively stimulating regional extracortical osteoclastic and endosteal osteoblastic activity, gives rise to slow, progressive, concentric diaphyseal atrophy/remodelling of the tubular bones of the hands and feet [11, 52, 54]. Detailed information regarding the bony changes indicative of autonomic peripheral neuropathy can be found in S6 Text.

In **KD271**, the hand and foot bones do not provide any skeletal evidence for the development of autonomic peripheral neuropathy. Nevertheless, in leprosy, it is an invariable accompaniment of impaired cutaneous and/or deep tissue sensation [53]. Therefore, based on the observed bony changes indicative of combined sensory and motor dysfunction of the left and right ulnar and posterior tibial nerves, the presence of autonomic peripheral neuropathy can be presumed in all of the four limbs of the middle-aged male.

In the right foot of **KD520**, there is direct evidence for the development of autonomic peripheral neuropathy as some of the middle-aged female's right foot bones displayed concentric diaphyseal atrophy [52, 54]. The right $2^{nd}$, $3^{rd}$, and $4^{th}$ metatarsals represented the knife-edge type, whereas the right $5^{th}$ proximal phalanx represented the true pan-circumferential concentric type (Fig 10) [20]. Although the two extant left metatarsals and the only observable left phalanx did not show any sign of concentric diaphyseal atrophy; and therefore, there is no skeletal evidence for the presence of autonomic dysfunction in the left foot, it is permissible to infer it. This is because loss of sensation, that can be presumed in the left foot of **KD520**, is always accompanied by autonomic impairment [53]. Unfortunately, based only on the middle-aged female's observable bone remains, it cannot be determined if her phalanges are missing *ante-* or *post-mortem* (only the right $5^{th}$ and an undeterminable left proximal phalanx are extant). Even if all of the right foot phalanges of **KD520** were present at death, the progressive concentric diaphyseal atrophy of the $2^{nd}$, $3^{rd}$, and $4^{th}$ right metatarsals and, at least, the right $5^{th}$ proximal phalanx, as well as the accompanying ulceration could have resulted in foot disfigurement and disability [12]. It should be noted that the same conclusion cannot be drawn for the left lower limb as the majority of its metatarsals and phalanges are missing. However, similar to the right foot, the left foot could have also become disfigured and disabled. This is because even its ulceration alone could have given rise to these severe biological consequences [94]. Dysfunction of the small cutaneous autonomic nerve fibres in the feet of **KD520** could have led to anhidrosis of the anaesthetic skin patches [76, 96, 97]. Due to the impaired or diminished sweat gland function, her anhidrotic skin could have become xerotic and hyperkeratotic, and thereby could have more readily cracked even with normal usage [76, 96, 97]. This could have made it even more susceptible to ulceration and secondary pyogenic infections [76, 96, 97]. Delayed ulcer healing can also be a consequence of peripheral autonomic impairment [96, 97]. Damage to the vascular autonomic innervation in the affected skin patches could have resulted in loss of vascular tone and subsequent stasis of the capillary blood flow with impaired healing of ulceration in the middle-aged female's both feet [96, 97]. In leprosy patients, non-healing plantar ulcers are a common cause of disability and consequently of reduced quality of life today [98, 99].

It was suggested that concentric diaphyseal atrophy of the hand and/or foot bones is virtually pathognomonic for leprosy. However, in cases where no other bony changes suggestive of Hansen's disease can be observed in other areas of the skeleton (e.g., the rhinomaxillary region of the face), as in **KD520**, other aetiologies should also be considered in the differential diagnosis. The most relevant ones are certain types of erosive arthropathies (e.g., rheumatoid arthritis

and psoriatic arthritis), hereditary sensory and autonomic neuropathies, and *diabetes mellitus* (S7 Text) [95].

**Osteoporotic bony changes indicative of disuse.** In leprosy patients, multiple factors (e.g., paralysis and deformity) can result in restricted mobility and consequent development of disuse osteoporosis in the affected limb(s) [20]. Inactivity-induced imbalance in the osteoclastic/osteoblastic equilibrium leads to disuse atrophy with consequent thinning of the cortical bone and decrease in density of the cancellous bone [20, 21, 52].

During the macromorphological examination of the skeletal remains of **KD271** and **KD520**, no radiological investigations could be conducted; and therefore, the assessment of the limb bones regarding the signs of osteoporosis was very limited. Nonetheless, no evident macromorphological signs of osteoporosis (e.g., substantially reduced cortical thickness, fragility fractures or unusually lightweight bones [11, 20, 52]) could be detected with the naked eye in the two presented cases (**KD271** and **KD520**) from the 7th-century-CE cemetery of Kiskundorozsma–Daruhalom-dűlő II.

## Conclusions

Although a number of differential diagnoses need to be considered for the individual bony changes observed in the skeleton of **KD271** and **KD520**, based on the nature, association, and distribution pattern of the registered alterations, they are most consistent with Hansen's disease in both individuals.

The middle-aged male (**KD271**) could have suffered from an advanced-stage, long-standing near-lepromatous or lepromatous form [11, 19, 49, 53]. The aDNA analysis has confirmed the macromorphologically established diagnosis as the sample from the hard palate of **KD271** appeared to be *M. leprae* DNA-positive [14, 25]. The disease has affected not only the rhinomaxillary region but also both upper and lower limbs. It has resulted in severe deformation and disfigurement of the aforementioned anatomical areas (saddle-nose, claw-hand, and flat-foot deformity, respectively) with his inability to perform the basic activities of daily living, such as eating, drinking, grasping, standing or walking.

On the other hand, the middle-aged female (**KD520**) can indicate the other extreme of Hansen's disease. Based on the absence of rhinomaxillary lesions in her skeletal remains, she could have suffered from a near-tuberculoid or tuberculoid form [22, 24]. It should be noted that the diagnosis of leprosy is weaker in **KD520** than in **KD271**, as the aDNA analysis gave negative results; and thus, could not strengthen the macromorphologically established diagnosis in the middle-aged female's case [28]. Nevertheless, in consideration that near-tuberculoid and tuberculoid leprosy are paucibacillary forms of Hansen's disease and the skeletal lesions indicative of leprosy in **KD520** could not be secondary to direct invasion of the affected bones by leprosy bacilli but to sensory and autonomic peripheral neuropathy, it is not surprising that the pathogen DNA could not be detected in the middle-aged female's skeletal remains [100]. Furthermore, as an adage says: "absence of evidence is not evidence of absence" [101]. Thus, even if the aDNA analysis gave negative results, based on the detailed differential diagnosis of concentric diaphyseal atrophy (S7 Text), its association with inflammatory bony changes of the foot and lower leg bones, as well as the high number of leprosy cases in the Kiskundorozsma–Daruhalom-dűlő II osteoarchaeological series, tuberculoid leprosy seems to be the most likely diagnosis in **KD520**. Similar to **KD271**, Hansen's disease could have resulted in disfigurement and disability of both of the lower limbs of **KD520**. Thus, the middle-aged female would have experienced difficulties in standing and walking, and thereby in conducting domestic and occupational physical activities.

KD271 and KD520 are amongst the very few leprosy cases that have already been published from the Avar Age of the Hungarian Duna–Tisza Interfluve, and the only examples where the detailed macromorphological description of the observed bony changes and their differential diagnoses have been provided. These two demonstrated cases from the Kiskundorozsma–Daruhalom-dűlő II archaeological site, representing the two extremes of Hansen's disease (lepromatous form and tuberculoid form, respectively), give us a unique insight into the different manifestations of leprosy and their biological consequences. Furthermore, the cases of these two severely disabled individuals, especially of KD271 –who would have required regular and substantial care from others to survive–imply that in the Avar Age community of Kiskundorozsma–Daruhalom-dűlő II there was a willingness to care for people in need.

## Supporting information

**S1 Text. The Avar khaganate in the carpathian basin between the 6[th] and 8[th] centuries CE.**
(PDF)

**S2 Text. Components of the rhinomaxillary syndrome in leprosy.**
(PDF)

**S3 Text. Differential diagnoses of the rhinomaxillary bony changes indicative of leprosy that were observed in KD271.**
(PDF)

**S4 Text. Non-specific bony changes indicative of motor peripheral neuropathy in leprosy.**
(PDF)

**S5 Text. Non-specific bony changes indicative of sensory peripheral neuropathy in leprosy.**
(PDF)

**S6 Text. Non-specific bony changes indicative of autonomic peripheral neuropathy in leprosy.**
(PDF)

**S7 Text. Differential diagnoses of the postcranial bony changes indicative of leprous autonomic peripheral neuropathy that were observed in KD520.**
(PDF)

**S1 Fig. Demographic profile of the Avar Age cemetery of Kiskundorozsma–Daruhalom-dűlő II (n = 94).**
(PDF)

## Author Contributions

**Conceptualization:** Olga Spekker.

**Data curation:** Olga Spekker.

**Funding acquisition:** Olga Spekker, Tivadar Vida, György Pálfi.

**Investigation:** Olga Spekker, Balázs Tihanyi, Orsolya Anna Váradi, Antónia Marcsik, Erika Molnár.

**Methodology:** Olga Spekker.

**Project administration:** Olga Spekker.

**Resources:** György Pálfi.

**Supervision:** Helen D. Donoghue, David E. Minnikin, Csaba Szalontai, Tivadar Vida, György Pálfi, Antónia Marcsik, Erika Molnár.

**Visualization:** Olga Spekker, Balázs Tihanyi, Luca Kis.

**Writing – original draft:** Olga Spekker, Tivadar Vida.

**Writing – review & editing:** Olga Spekker.

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
