## [Decision Letter · Decision Letter 0]

29 Sep 2021

PONE-D-21-21682The two extremes of Hansen’s disease – Different manifestations of leprosy and their biological consequences in an Avar Age (late 7th century CE) osteoarchaeological series of the Duna-Tisza Interfluve (Kiskundorozsma–Daruhalom-dűlő II, Hungary)PLOS ONE

Dear Dr. Spekker,

Thank you for submitting your manuscript to PLOS ONE. After careful consideration, we feel that it has merit but does not fully meet PLOS ONE’s publication criteria as it currently stands. Therefore, we invite you to submit a revised version of the manuscript that addresses the points raised during the review process.

I now have two expert reviews of this work. Thank you for your patience as we worked to review your manuscript. Both reviewers felt there was merit in the work. There were suggestions for additional experimentation if possible as well as extensive edits. This is considered a major revision.

We look forward to receiving your revised manuscript.

Kind regards,

JJ Cray Jr., Ph.D.

Academic Editor

PLOS ONE

Journal Requirements:

2. In your manuscript, please provide additional information regarding the specimens used in your study. Ensure that you have reported specimen numbers and complete repository information, including museum name and geographic location.

For more information on PLOS ONE's requirements for paleontology and archaeology research, see https://journals.plos.org/plosone/s/submission-guidelines#loc-paleontology-and-archaeology-research.

5. We note that Figures 2 - 11 in your submission contain copyrighted images. All PLOS content is published under the Creative Commons Attribution License (CC BY 4.0), which means that the manuscript, images, and Supporting Information files will be freely available online, and any third party is permitted to access, download, copy, distribute, and use these materials in any way, even commercially, with proper attribution. For more information, see our copyright guidelines: http://journals.plos.org/plosone/s/licenses-and-copyright.

a. You may seek permission from the original copyright holder of Figures 2 - 11 to publish the content specifically under the CC BY 4.0 license. 

Reviewers' comments:

Reviewer's Responses to Questions

**Comments to the Author**

1. Is the manuscript technically sound, and do the data support the conclusions?

Reviewer #1: Yes

Reviewer #2: Partly

2. Has the statistical analysis been performed appropriately and rigorously? 

Reviewer #1: Yes

Reviewer #2: N/A

3. Have the authors made all data underlying the findings in their manuscript fully available?

Reviewer #1: Yes

Reviewer #2: Yes

4. Is the manuscript presented in an intelligible fashion and written in standard English?

Reviewer #1: Yes

Reviewer #2: Yes

5. Review Comments to the Author

Reviewer #1: This is an interesting article that provides very useful information on the extreme/polar clinical manifestations (lepromatous leprosy and tuberculoid leprosy) of Hansen disease in skeletal samples and the corresponding biological consequences observed in two individuals from the 7th century-CE cemetery of Kiskundorozsma–Daruhalom-dűlő II, Hungary. The paleopathological analysis of both individuals was complemented by ancient DNA analysis (previously published), where one sample yielded a possible DNA-positive for Mycobacterium leprae.

Below, I provide my questions or recommendations for the authors:

1) I suggest choosing a single nomenclature for the disease, either Hansen disease or leprosy (or explain why using both terminologies). For example, in Results, the authors use HD (main text) but in most figure legends they use leprosy. If historical context is no needed or necessary, I prefer Hansen disease or (if the authors agree or if necessary) the more “neutral” term mycobacterial neurodermatosis.

2) I do not understand why the authors selected to include in Discussion a long general introduction of each category of the pathognomonic bone changes (bony changes of the skull; postcranial skeleton; leprosy-specific boney changes, and non-specific bony changes…). Yes, I agree that all this introductory information is important and useful (and the authors did an excellent summary for all of them), but most general information should be in the Introduction of the article, not the Discussion. The Discussion should be based mostly on the findings/conclusions observed on the two individuals (KD217 and KD520). I agree on adding (or repeating) in Discussion some useful osteological information to give context to the findings in this report.

The authors could keep same subtitles in Discussion (Bony changes of the Skull; Bony changes of the postcranial skeleton…) but the text should be mainly focused on the Results; if the authors agree, the long introductory sections (most of the text/description) for bony changes could fit better in the Introduction.

As mentioned before, following the title "The two extremes of Hansen's disease...", the authors should leave or bring some general osteological information to explain or justify why KD217 could be associated with LL, and KD520 with TL (i.e., as described for KD271, on page 20, ln 481-482, the perforation of the hard palate is a late phenomenon of LL)

3) I consider important the recognition of TL in skeletal samples (as done by V. Matos and other colleagues), and the authors (sorry if I am confused) initially presented in the Discussion that the evidence for concentric diaphyseal atrophy (right foot) is crucial to associate KD520 with TL, but at the end (pg. 35, ln 847-848) the authors cautiously wrote that “…as in KD520, other aetiologies should also be considered in the differential diagnosis”.

Do the authors conclude that KD520 could be associated with TL because not only concentric diaphyseal atrophy was observed but inflammatory lesions (bilateral) on tibial nerve as well?

If my observation is correct, I recommend expanding on that (if not, it looks that the TL diagnose for KD520 is weak or not definitive)

The article is clear and well written, and I do celebrate new studies on Hansen disease in ancient skeletal samples that attempt to recognize osteological differences between LL and TL polar states. My recommendation is that the article should undergo minor revisions before final approval/acceptance.

Reviewer #2: Dear authors,

This manuscript presents two interesting paleopathological case studies from an Hungarian archaeological context from 7th century AD.

GENERAL COMMENTS:

The diagnosis of lepromatous leprosy for individuals KD271 is very convincing. Moreover, this diagnosis is supported by previously published paleomicrobiological analysis.

The diagnosis of tuberculoid leprosy in skeleton KD520 is, in my opinion, more complex. I think it would have been useful to present any DNA results that might already exist for this skeleton. It is suggested to clarify this. Was skeleton KD520 tested for aDNA of M. leprae? If yes, what were the results?

KEYWORDS:

- please consider replacing “type” by “leprosy” (eg. tuberculoid leprosy)

ABSTRACT:

- add Hungary (l. 29)

- add the total number of skeletons (n=94) from this archaeological series

INTRODUCTION:

- the text between lines 57-73 is really useful for the paper purposes? please consider removing it

- please check the spelling of “achro-osteolysis”. Is it “acro-osteolysis”? (line 129 and bellow)

- please consider replacing: “hand and/or foot bones” by “hand and/or foot phalanges, metacarpals and metatarsals” (line 130)

- Objectives are clear and relevant.

M&M:

- please add the specific methods used to estimate sex and age at death of skeletons KD271 and KD520. It is unclear how these parameters were estimated for these individuals. (lines 247-250)

- It is recommended adding the overall age and sex profiles of the skeletal series, if possible

RESULTS:

- the two cases are described in detail and bony lesions are very well described and shown in the figures.

- It is suggested to add a photo (to be included in Fig. 4) showing the nasal surface of the palatine process of sk. 271.

- please add the dimensions of the perforation of the palatine process (line 273)

- please consider replacing: “tarsal bones of the feet” by “tarsal bones” (line 307)

DISCUSSION:

- This section is, in my opinion, very extensive and needs improvements including text reduction.

- Please consider removing the following text: lines 416-453, 543-574, 625-690, 771-808. These parts are not properly discussion and are not necessary to discuss the two

- cases.

- it is suggested removing the word “pathognomonic” in line 404

- please consider deleting “of the face” in line 456

- it is suggested to avoid the term periostitis. Alternative: periosteal new bone

- about osteoporosis (line 859): what methods were applied to evaluate osteoporosis? this is unclear and needs clarification

- The discussion of the “Non-specific bony changes…” was made by dividing in three parts: motor, sensory, and autonomic peripheral neuropathy. I’m not sure if this “classic” artificial division, influenced by the radiological literature, is the most useful one. The authors are invited to rethink the organization and text reduction of this part of the discussion.

Best reagrds.

6. PLOS authors have the option to publish the peer review history of their article (what does this mean?). If published, this will include your full peer review and any attached files.

Reviewer #1: No

Reviewer #2: No

---

## [Editor Report · Decision Letter 1]

10 Nov 2021

PONE-D-21-21682R1

The two extremes of Hansen’s disease – Different manifestations of leprosy and their biological consequences in an Avar Age (late 7th century CE) osteoarchaeological series of the Duna-Tisza Interfluve (Kiskundorozsma–Daruhalom-dűlő II, Hungary)

Dr. Olga Spekker

Dear Dr. Spekker,

Thank you for reaching out and bringing this issue to our attention.

In response to your request, we will reinstate your submission and resume the evaluation of your work, taking into account your response to the original concerns raised.

 I hope that I can help explain the current status of your manuscript. 

Firstly, accept my apologies for the error that occurred with your manuscript. Please know that there was nothing that the authors had done to cause your paper to be withdrawn, and this was an error on behalf of the journal. Your manuscript was unfortunately misattributed as resubmission to a different manuscript, and therefore, your paper was withdrawn in error.

The manuscript has been returned to the 'Submissions Needing Revision' folder in your Editorial Manager account. If you would like to incorporate any changes to the manuscript in response to the original evaluation, please complete your revisions and resubmit your manuscript files. 

Your resubmission is due on Dec 25 2021 11:59PM. 

The manuscript will undergo another editorial evaluation, which may involve further external review at the editors' discretion, we thus cannot anticipate the outcome of this review process. Please note that the editorial decision reached following consideration final but you will have the opportunity to revise or resubmit if you wish.

Please do not hesitate to contact us at plosone@plos.org if you have any queries regarding the appeal process.

With best wishes, 

Alexis Miller

Staff

PLOS ONE

---

## [Author Response · Author response to Decision Letter 1]

11 Nov 2021

Dear Dr JJ Cray Jr.,

I am very thankful for the reviewers’ insightful and constructive comments regarding our manuscript entitled “The two extremes of Hansen’s disease – Different manifestations of leprosy and their biological consequences in an Avar Age (late 7th century CE) osteoarchaeological series of the Duna-Tisza Interfluve (Kiskundorozsma–Daruhalom-dűlő II, Hungary)” that was submitted to the PLOS ONE (manuscript ID: PONE-D-21-21682). I am sure that the reviewers helped us to improve the quality of our manuscript. The main text has been modified following the reviewers’ suggestions. Furthermore, some of the figures were replaced to avoid copyright issues that were mentioned in the editorial letter. The revised files have been uploaded to the submission site of the PLOS ONE.

Responses to the suggestions:

1) Reviewer 1 commented that in the manuscript, we should explain why we use both “Hansen’s disease” and “leprosy” throughout the text. In Reviewer 1’s opinion, we should use a single nomenclature for the disease – “Hansen’s disease”, “leprosy”, or “mycobacterial neurodermatosis”. We agree with Reviewer 1 that “mycobacterial neurodermatosis” is a more neutral term, and considering the stigma and prejudice that is related to leprosy, “mycobacterial neurodermatosis” could be an appropriate choice instead of “leprosy”. However, to the best of our knowledge, in the palaeopathological literature, “leprosy” is the most frequently used nomenclature for the disease. Recently, “Hansen’s disease” started to be in use but unfortunately, we have never seen the term “mycobacterial neurodermatosis” in palaeopathological papers. In most of the articles we could find in the palaeopathological literature, “Hansen’s disease” was mentioned as another name for leprosy but then the term “leprosy” was used throughout the text. In the minority of the palaeopathological papers, both “leprosy” and “Hansen’s disease” were used throughout the text. As, in our opinion, palaeopathologists would represent the main readership of our paper, we would like to keep the term “leprosy”, as this term would be the most familiar for them. If Reviewer 1 agrees, we would like to use the term “Hansen’s disease”, as well. On the one hand, by using both terms, we would like to avoid word repetition, especially in sentences where the name of the disease would be mentioned twice. On the other hand, by using not only “leprosy” but “Hansen’s disease”, we would like to contribute to this term to become more frequently used in the palaeopathological literature. We hope that by defining “leprosy” and “Hansen’s disease’ as interchangeable names of the same disease in the first sentence of our manuscript, we can avoid confusion in the reader that could derive from the usage of both terms.

2) Reviewer 1 noted that “the long introductory sections (most of the text/description) for bony changes could fit better in the Introduction” instead of the Discussion. Reviewer 2 also mentioned that we should consider removing “lines 416-453, 543-574, 625-690, and 771-808” from the Discussion as “these parts are not properly discussion and are not necessary to discuss the two cases”. We agree with the reviewers that the Discussion was not the best choice for these parts to be placed but we did not want to put these long texts into the Introduction because we thought that it would have become too long. On the other hand, we thought that by putting this general information into the Discussion, right before the detailed discussion of the detected bony changes and their biological consequences, it would make it easier for the reader to connect the medical and palaeopathological literature data to the bony changes observed in KD271 and KD520. We also agree with Reviewer 1 that the aforementioned introductory information is important and useful, especially if the reader is not an expert in the palaeopathological diagnosis of leprosy. Therefore, instead of completely deleting these parts from our manuscript, following the reviewers’ advice, we removed this introductory information from the Discussion and put it into the Supporting Information (S2, S4, S5, and S6 Texts). This way, the Introduction did not become annoyingly long and the introductory general information from the Discussion remained available for the reader. As the replacement of these paragraphs changed the order of some references, they were renumbered accordingly (both in the main text and reference list). Furthermore, following Reviewer 1’s suggestion, we kept the subtitles in the Discussion and left “some general osteological information to explain or justify why KD271 could be associated with LL, and KD520 with TL”.

3) Reviewer 1 commented that we should expand our argumentation why we think that in KD520, TL is the most likely diagnosis, as based on the original version of our argumentation, it seems that the diagnosis of TL for KD520 is weak or not definitive. We agree with Reviewer 1 that we should expand this aspect in the revised version of our manuscript, as we only briefly mentioned in the Conclusions part of the original version that “Although a number of differential diagnoses need to be considered for the individual bony changes observed in the skeleton of KD271 and KD520, based on the nature, association, and distribution pattern of the registered alterations, they are most consistent with HD in both individuals.”. Even in KD271, where not only the postcranial skeleton but also the rhinomaxillary region seemed to be affected by leprosy, we had to provide a detailed differential diagnosis of the observed rhinomaxillary bony changes (S2 Text in the original version) to exclude the other possible aetiologies (e.g., syphilis or tuberculosis). In the case of KD271, we were lucky that the diagnosis of HD could be established based not only on the observed macromorphological bony changes and their association with each other but on the results of the aDNA analysis, as the sample from the hard palate appeared to be M. leprae DNA-positive. We also agree with both reviewers that the diagnosis of TL in osteoarchaeological series is more complex than that of LL – in the palaeopathological literature, there have been only a few attempts to identify TL cases. Based on the above, in the revised version of our manuscript, we highlighted that the diagnosis of HD is weaker in KD520 than in KD271. We agree with Reviewer 2 that a positive aDNA result could strengthen the diagnosis of leprosy in KD520. However, the chance for successfully detecting M. leprae DNA in historic TL cases is very low. On the one hand, TL is a paucibacillary form of Hansen’s disease (in contrast to LL, which is a multibacillary form). On the other hand, the (postcranial) bony changes developing in TL cases, such as KD520, are not secondary to direct invasion of the affected bones by leprosy bacilli but to peripheral neuropathy (in contrast to the rhinomaxillary changes observed in LL cases, such as KD271, that directly result from the presence of the pathogens in the affected bones). Therefore, it is not surprising that the aDNA analysis for M. leprae gave negative results in KD520 as very likely, leprosy bacilli have not been present in her bones even when she was alive (reference 100 in the revised manuscript). Furthermore, as an adage in aDNA says: “absence of evidence is not evidence of absence” (reference 101 in the revised manuscript). Thus, even if the aDNA analysis could not strengthen the macromorphological diagnosis of TL in KD520, in our opinion, based on the detailed differential diagnosis (S3 Text in the original version of our manuscript), in which we excluded the other possible aetiologies, the macromorphological appearance and association of the observed bony changes (i.e., concentric diaphyseal atrophy and inflammatory bony changes of the small bones of the feet, and the tibiae and fibulae), as well as the high number of leprosy cases in the Kiskundorozsma–Daruhalom-dűlő II osteoarchaeological series (among which three proved to be M. leprae DNA-positive), TL seems to be the most likely diagnosis in KD520. Following Reviewer 1’s advice, this aspect was expanded in the revised version of our manuscript. Because of this, two extra references (references 100 and 101) were added to the manuscript, both in the main text and reference list. It seems that our phrasing in the original manuscript has been unclear, as we already mentioned in that version that an aDNA analysis for M. leprae was performed not only on KD271 but also on KD520. However, as it could have been expected, it gave positive results only in KD271 (lines 255–259 and 877–878 in the original version of the manuscript). We are really sorry for the misunderstanding generated by our inappropriate phrasing in the original version of the manuscript. Following Reviewer 2’s suggestion, to clarify that there was an aDNA analysis for M. leprae on KD520 and it gave negative results, the relevant sentence was supplemented in the Conclusions.

4) Following Reviewer 2’s suggestion, “type” was replaced by “leprosy” in the Keywords (“lepromatous type” was changed to “lepromatous leprosy” and “tuberculoid type” was changed to “tuberculoid leprosy”).

5) Following Reviewer 2’s comment, “Hungary” and the total number of skeletons from the archaeological site of Kiskundorozsma-Daruhalom-dűlő (“n=94”) was added in line 29 in the Abstract.

6) Reviewer 2 noted that we should consider removing the text between lines 57–73 (in the original version of our manuscript). Following Reviewer 2’s suggestion, some of the sentences between lines 57–73 were deleted. Nonetheless, some of the sentences between lines 57–73 were kept, as in our opinion, they contain important information that are later referred in the text or necessary for the reader to understand the following paragraphs in the Introduction, Discussion, and Supporting Information parts of the manuscript (e.g., the main portal of entry and exit for leprosy bacilli is the nasal mucosa; not all the infected individuals develop active disease and consequently, bony changes due to genetic susceptibility; and leprosy has a long incubation period and in some cases, the infection can spontaneously resolve).

7) Reviewer 2 suggested that we should check the spelling of “achro-osteolysis”. In a number of papers we used as references (e.g., references 11, 20, 52, and 56 in the original version of our manuscript), it was spelled as “achro-osteolysis”. This is why we used this spelling. However, following Reviewer 2’s advice, the spelling was changed to “acro-osteolysis” throughout the text.

8) Following Reviewer 2’s comment, we replaced “hand and/or foot bones” by “hand and/or foot phalanges, metacarpals, and metatarsals” in line 130 (in the original version of our manuscript).

9) Reviewer 2 mentioned that we should “add the specific methods used to estimate sex and age at death of skeletons KD271 and KD520”, because “it is unclear how these parameters were estimated for these individuals (lines 247–250)”. We agree with Reviewer 2 that, as we put all the references of the applied methods to the end of the relevant sentence in lines 247–250 of the original manuscript, we did not specify, which methods were used for age at death estimation and which ones for sex determination. To clarify this, we put the appropriate references after “age at death” and “sex” in the relevant sentence in the revised version: “Age at death [36-45] was estimated and sex [46] was determined applying standard macromorphological methods of biological anthropology.” As the order of the references changed, we renumbered them (both in the main text and reference list).

10) Following Reviewer 2’s advice, a diagram presenting the demographic profile of the examined skeletal series was added as Supplementary figure 1.

11) Reviewer 2 asked us to “add a photo (to be included in Fig. 4) showing the nasal surface of the palatine process of sk. 271” and “add the dimensions of the perforation of the palatine process (line 273)”. Unfortunately, we cannot provide such a photo and cannot add the dimensions of the perforation. As it has been mentioned in the original version of our manuscript, an aDNA analysis was performed on KD271 and the sample was taken from the hard palate; consequently, the majority of the hard palate was destroyed. Therefore, during the macromorphological re-evaluation of KD271, we could not take photos of the nasal and oral surfaces of the maxillary palatine process and could not measure the perforation of the hard palate. To describe the lesions of the nasal and oral surfaces of the hard palate, we used the original, very short description of KD271 that was provided by Marcsik and her co-workers in 2007 (reference 28 in the original version of our manuscript) and the photos that were taken by them back in the 2000s. Unfortunately, in the original description, there was no information regarding the dimensions of the perforation, and among the original photos taken by Marcsik and her colleagues in the 2000s, Figures 4E and 4F were the only ones which we could use in our manuscript to present the oral and nasal surfaces of the hard palate, respectively. This is why we highlighted in the legend of Figure 4 that “Photos A–D were taken after sampling for aDNA analysis, whereas photos E–F were taken before sampling for aDNA analysis.”. Nevertheless, in our opinion, the scale on Figure 4E can help the reader to at least estimate the dimensions of the perforation of the hard palate. Since, based on this scale, it would not be an accurate way to measure the exact dimensions of the lesion, if Reviewer 2 agrees, we would not include these estimates to the manuscript.

12) Following Reviewer 2’s suggestion, “tarsal bones of the feet” was replaced by “tarsal bones” in line 307 (in the original version of our manuscript), “pathognomonic” was removed from line 404, and “of the face” was deleted from line 456 (in the original version of our manuscript).

13) Reviewer 2 asked us to avoid the term “periostitis” and use “periosteal new bone” instead. Unfortunately, we cannot avoid the term “periostitis”, as in the sentences we used it, we wanted to refer to the condition, i.e., inflammation of the periosteum, that results in deposition of new bone on the extracortical surface of the affected bone subjacent to the inflamed periosteum. In those sentences, where we wanted to refer to the signs of periostitis, i.e., the periosteal new bone depositions, that could have been observed on the skeletal remains of both KD271 and KD520, we used the term “subperiosteal new bone formation” as in those cases, we wanted to refer not to the condition leading to the development of the detected subperiosteal new bone formations but the bony changes themselves. Throughout the manuscript, we tried to be very cautious about when we referred to the condition, i.e., periostitis, and when to its bony signs, i.e., subperiosteal new bone formations. Therefore, if Reviewer 2 agrees, we would like to use the term “periostitis” instead of “periosteal new bone” in the relevant sentences.

14) Reviewer 2 asked us to mention “what methods were applied to evaluate osteoporosis” as “it is unclear and needs clarification”. Unfortunately, during the macromorphological re-evaluation of the skeletal remains of KD271 and KD520, they could be examined only with the naked eye (no radiological investigations could be conducted). Therefore, the assessment of the affected limb bones regarding the signs of osteoporosis was very limited. Nevertheless, no evident macromorphological signs of osteoporosis (e.g., substantially reduced cortical thickness, fragility fractures or unusually lightweight bones) could be detected in the affected limb bones of KD271 and KD520 (e.g., references 11, 20, and 52 in the original version of the manuscript). To further highlight this aspect, it would be very useful if a detailed radiological examination could also be performed on the skeletal remains of KD271 and KD520 in the future. Following Reviewer 2’s comment, this part of the Discussion was supplemented.

15) Regarding the discussion of the non-specific bony changes that was made by dividing in three parts (motor, sensory, and autonomic peripheral neuropathy), Reviewer 2 noted that “I’m not sure if this “classic” artificial division, influenced by the radiological literature, is the most useful one.” We chose this division because in our opinion, it is the most appropriate one to present the association between the macromorphological bony changes that were detected with the naked eye in the skeleton of KD271 and KD520 and the exact aetiology, i.e., sensory, motor, and autonomic peripheral neuropathy, that led to the development of the observed alterations. For this reason, if Reviewer 2 agrees, we would like to keep this division.

16) It was stated in the editorial letter that we should “provide additional information regarding the specimens used in your study. Ensure that you have reported specimen numbers and complete repository information, including museum name and geographic location”. Following this comment, in the Ethics Statement, the specimen numbers (KD271 and KD520) were complemented by the inventory and grave numbers of the evaluated individuals, and the geographic location of the two skeletons (KD271 and KD520) was supplemented by the postal address of the Department of Biological Anthropology, University of Szeged (Szeged, Hungary).

17) In the editorial letter, it was stated that “in order to use the direct billing option the corresponding author must be affiliated with the chosen institute. Please either amend your manuscript to change the affiliation or corresponding author, or email us at plosone@plos.org with a request to remove this option”. We do not really understand why it seems that our corresponding author (Olga Spekker) is not affiliated with the chosen institute (University of Szeged). On the title page of the original version of our manuscript, both the University of Szeged and the Eötvös Loránd University were mentioned as her affiliations, since she has a job at both universities. In the online submission system, she named the University of Szeged as her affiliation, since she applied for funding to the Szeged Open Access Fund to cover the publication fees. Therefore, in our opinion, it is not necessary to change the corresponding author or to change the affiliation of the corresponding author. We hope that after rechecking of the affiliation of the corresponding author (Olga Spekker) in the manuscript file and online submission system, the Journal Office will not find it necessary to amend our manuscript (i.e., to change the corresponding author or the affiliation of the corresponding author).

18) It was stated in the editorial letter that some of the images in our figures may be copyrighted. To avoid copyright issues, Figures 1A, 2C, and 2D were replaced. Unfortunately, as we do not know the original source of the map and the drawing of a skeleton that were used after substantial modifications in Figures 1 and 2 in the original version of our manuscript, we cannot obtain permission from the original copyright holder. For the same reason, we do not know if the above-mentioned source images are actually copyrighted or not. That is why we decided to supply replacement figures in the revised version of our manuscript. It should be mentioned that in Figures 2C and 2D, the left tibia of KD271 and KD520 are missing from the replacement photos, whereas in the original image (drawing of a skeleton), they were marked as available for examination. This is because these bones were available for investigation back in the 2000s (this preliminary evaluation was performed by Antónia Marcsik and Erika Molnár, two of the co-authors of this manuscript) but later, have been transported abroad for further analyses. Unfortunately, the above-mentioned two tibiae have never been returned to the University of Szeged (Szeged, Hungary); therefore, they were not available for re-evaluation and photo-shooting, performed only a couple of months ago. For the same reason, similar to the bony changes of the hard palate of KD271, to describe the lesions of these two tibiae, we used the original descriptions that were provided by Marcsik and her co-workers in 2007 (reference 28 in the original version of our manuscript) and photos that were taken by them back in the 2000s. As they have not taken photos of the complete skeletons of KD271 and KD520 to present their completeness, we had to take new photos, but without the left tibiae, to use them as replacement images for Figures 2C and 2D that present the completeness of the two examined skeletons. As for Figures 3–11, most of the pictures were taken and edited by Luca Kis, one of the co-authors, for this manuscript right before the original submission; they have never been published elsewhere. Figures 4E and 4F were taken by another co-author, Erika Molnár, back in the 2000s, but they have never been published elsewhere either. Therefore, in our opinion, the aforementioned figures are not copyrighted and we can use them freely in our manuscript without providing a Content Permission Form. We hope that after rechecking our provided figures, the Journal Office will not find it necessary to replace Figures 3–11.

In the revised version of our manuscript, we tried to execute all suggestions of the reviewers. I hope this new version will be suitable for publication in the PLOS ONE.

Thank you again for the reviewers’ insightful and constructive comments and your editorial work!

Yours sincerely,

Dr Olga Spekker, PhD

corresponding author

---

## [Decision Letter · Decision Letter 2]

16 Feb 2022

PONE-D-21-21682R2The two extremes of Hansen’s disease – Different manifestations of leprosy and their biological consequences in an Avar Age (late 7th century CE) osteoarchaeological series of the Duna-Tisza Interfluve (Kiskundorozsma–Daruhalom-dűlő II, Hungary)PLOS ONE

Dear Dr. Spekker,

Thank you for submitting your manuscript to PLOS ONE. After careful consideration, we feel that it has merit but does not fully meet PLOS ONE’s publication criteria as it currently stands. Therefore, we invite you to submit a revised version of the manuscript that addresses the points raised during the review process.

There are some additional details and edits that should be able to be accomplished in a minor revision. Note only an editorial desk review will be conducted upon resubmission.

We look forward to receiving your revised manuscript.

Kind regards,

JJ Cray Jr., Ph.D.

Academic Editor

PLOS ONE

Journal Requirements:

Reviewers' comments:

Reviewer's Responses to Questions

**Comments to the Author**

1. If the authors have adequately addressed your comments raised in a previous round of review and you feel that this manuscript is now acceptable for publication, you may indicate that here to bypass the “Comments to the Author” section, enter your conflict of interest statement in the “Confidential to Editor” section, and submit your "Accept" recommendation.

Reviewer #3: All comments have been addressed

Reviewer #4: All comments have been addressed

2. Is the manuscript technically sound, and do the data support the conclusions?

Reviewer #3: Yes

Reviewer #4: Yes

3. Has the statistical analysis been performed appropriately and rigorously? 

Reviewer #3: I Don't Know

Reviewer #4: N/A

4. Have the authors made all data underlying the findings in their manuscript fully available?

Reviewer #3: Yes

Reviewer #4: Yes

5. Is the manuscript presented in an intelligible fashion and written in standard English?

Reviewer #3: No

Reviewer #4: Yes

6. Review Comments to the Author

Reviewer #3: I was not an original reviewer of the manuscript but I have read through the authors’ responses to the previous comments, which seem reasonable (though I should note too that I am not a specialist in paleopathology).

With that said, I do not see any reason to not use the term “leprosy”, which is commonly used in both scientific and laymen’s literature. While there may be a certain stigma attached to the term, it should not preclude its use in cases such as this.

I had a few other comments that I hope can help improve the paper, but will leave it to the previous reviewers to assess the quality of responses to revisions made.

Overall, the paper requires some editing and wordsmithing. For example, the sentence on lines 34-36 would be better written as: “The led to severe deformation and disfigurement of the involved anatomical areas of the skeleton, resulting in the inability to perform the basic…..”. There are quite a few places through the ms where things could be stated more simply.

The authors also use the “hook” of stating that the cases they report are: (lines 42-43) “amongst the very few published cases with leprosy from the Avar Age of the Hungarian Duna-Tisza Interfluve” (which is repeated a number of times in the paper). I’m sure this is the case, but it seems overly narrow to me. Without knowing much about the temporal range or geographical location of the specimens, it doesn’t seem that significant to me. Could the authors qualify this? Does this represent a period of centuries across a wide geographical area that was known to have significant outbreaks of leprosy but these are really the few that are known? If the chronological range and geographical area is narrow and limited, this statement just doesn’t seem that important on a global scale.

Lines 50 – better to say “infectious disease found mainly in humans”. With that said, I’m curious, is it also transmissible among other primates?

Line 56 – delete “of” before “caused”

Line 72 – replace “hardly” with “minimally”

P. 6 – I appreciate the historical background provided, but it would be helpful to provide a few more details about the transmission of leprosy to some other parts of the world to show just how pervasive, transmissible, and fearful it was to humans. A recent paper by Nelson et al. (2022) describe what is purported to be the earliest directly dated case in the Americas. There is also early evidence for the spread of leprosy to India (ca. 4000 BP) (see Robbins et al. 2009, 2013). These are just a few, but there are others that would be worth noting.

In addition, given some misunderstandings about leprosy and how it could be contracted, as well as the desire to help those afflicted, there were many leprosaria established in remote or protected areas (individual islands in the Caribbean and Pacific are well-known examples) to sequester and render aid.

Line 215 – replace “of” with “in biological anthropology”

Lines 217-218 – use semicolons to separate sequentially numbered orders

Line 360 – typo - should be “With” (not “Whit”)

Line 636 – consider revising to: “It is important to note that apart from HD….”

Line 639 and in several places after (e.g., Lines 670-674). There are a number of cases where longer or multiple sentences are in parentheses. These should usually be reserved for shorter explanations, so consider incorporating these into the text without parentheses.

Line 674 – replace “decided” with “determined”

Line 699 – remove double parentheses; instead, parentheses within parentheses should be changed to brackets (e.g., “….arthritis [RA] and psoriatic arthritis [PsA])

Line 738 – this is not an adage exclusive of aDNA, but is a commonly used phrase in science and beyond

Line 745 – better to say “….in standing and walking and various physical actitivies.”

Figure 1 is insufficient for an international audience. It is unclear where this region is located on a regional or global map. Please include one that shows where “A” is found. Map also needs a scale and north arrow.

Figures 2 and others – remove parentheses after letter, which is unnecessary

References

Nelson, G.C., Dodrill, T.N. and Fitzpatrick, S.M., 2022. A probable case of leprosy from colonial period St. Vincent and the Grenadines, Southeastern Caribbean. International Journal of Paleopathology, 36, pp.7-13.

Robbins, G., Tripathy, V.M., Misra, V.N., Mohanty, R.K., Shinde, V.S., Gray, K.M. and Schug, M.D., 2009. Ancient skeletal evidence for leprosy in India (2000 BC). PloS one, 4(5), p.e5669.

Robbins Schug, G., Blevins, K.E., Cox, B., Gray, K. and Mushrif-Tripathy, V., 2013. Infection, disease, and biosocial processes at the end of the Indus Civilization. PLoS One, 8(12), p.e84814.

Reviewer #4: My only comments are minor. They pertain to English and anatomical nomenclature. In the line 185"prevention" should be "salvage". In lines 275, 288 and 290 the term "turbinate" is used incorrectly. In humans the inferior concha is a separate bone, different from animal "turbinate" bones. A basic anatomical terminology error. Use inferior nasal concha.

In the era of electronic (digital, on-line) publications, the use of abbreviations/acronyms saving the cost of printing ink and paper for numerous copies of printed papers saves nothig while it appears to be pretentiously "scientific" and makes reading of manuscripts difficult. I recommend to use full terms instead of pretentious acronymes like TT, LL, HD etc.

7. PLOS authors have the option to publish the peer review history of their article (what does this mean?). If published, this will include your full peer review and any attached files.

Reviewer #3: No

Reviewer #4: **Yes: **Emeritus Professor Maciej Henneberg

---

## [Author Response · Author response to Decision Letter 2]

17 Feb 2022

Dr JJ Cray Jr., PhD

Academic Editor

PLOS ONE

February 17, 2022

Dear Dr JJ Cray Jr.,

I am very thankful for the reviewers’ insightful and constructive comments regarding our manuscript entitled “The two extremes of Hansen’s disease – Different manifestations of leprosy and their biological consequences in an Avar Age (late 7th century CE) osteoarchaeological series of the Duna-Tisza Interfluve (Kiskundorozsma–Daruhalom-dűlő II, Hungary)” that was submitted to the PLOS ONE (manuscript ID: PONE-D-21-21682). I am sure that the reviewers’ comments helped us to improve the quality of our manuscript. The main text, figures, and supplementary texts have been modified following the reviewers’ suggestions. The revised files have been uploaded to the submission site of the PLOS ONE.

Responses to the suggestions:

1) Reviewer 3 commented that “Overall, the paper requires some editing and wordsmithing. For example, the sentence on lines 34-36 would be better written as: “The led to severe deformation and disfigurement of the involved anatomical areas of the skeleton, resulting in the inability to perform the basic…..”. There are quite a few places through the ms where things could be stated more simply.” Following Reviewer 3’s advice, the aforementioned sentence was amended. We also made all the corrections in the main text that Reviewer 3 suggested in their review.

2) Reviewer 3 noted that “The authors also use the “hook” of stating that the cases they report are: (lines 42-43) “amongst the very few published cases with leprosy from the Avar Age of the Hungarian Duna-Tisza Interfluve” (which is repeated a number of times in the paper). I’m sure this is the case, but it seems overly narrow to me. Without knowing much about the temporal range or geographical location of the specimens, it doesn’t seem that significant to me. Could the authors qualify this? Does this represent a period of centuries across a wide geographical area that was known to have significant outbreaks of leprosy but these are really the few that are known? If the chronological range and geographical area is narrow and limited, this statement just doesn’t seem that important on a global scale.” Unfortunately, we do not agree with Reviewer 3 that “if the chronological range and geographical area is narrow and limited” the identification of leprosy cases from the Avar Age of the Duna-Tisza Interfluve (Hungary), like KD271 and KD520, cannot “seem important on a global scale”. As we already highlighted in our manuscript, “(sub)genotyping of M. leprae aDNA, reconstructed from ancient human cases with Hansen’s disease, provides invaluable information about the origins and geographical distribution of leprosy bacilli, and the migration routes of their human host over time” (lines 135–138 in the revised version of our manuscript). “Recent subgenotypic data have revealed that Europe could be a key for the early spread and global dissemination of Hansen’s disease” (lines 139–141 in the revised version of our manuscript). In addition, “it is suggested that after the fall of the Roman Empire, the successive westward migration of the nomadic Avar tribes from Central Asia or Asia Minor via the Middle East led to the separate introduction or re-transmission of different M. leprae strains into Eastern and Central Europe (including Hungary) during the early mediaeval period” (lines 147–151 in the revised version of our manuscript). We also highlighted in the S1 Text (entitled “The Avar Khaganate in the Carpathian Basin between the 6th and 8th centuries CE”) that “the concentration of high-status burials lavishly furnished with weapons and gold artefacts in the Duna-Tisza Interfluve suggests that the Avar power centre lay in this region during the 6th and 8th centuries.” Based on the above, this geographical area of Hungary was exceptionally important in the Avar Age (6th–8th centuries CE), when leprosy could be introduced or re-transmitted to Europe (including Hungary). However, as we stated in our manuscript, the number of identified leprosy cases from the Duna-Tisza Interfluve of Hungary is very low – besides KD271 and KD520 that are discussed in our current paper, two other cases from the archaeological site of Hajós–Cifrahegy have been published in English (Marcsik et al. 2016 – reference 29 in the revised version of our manuscript). In addition, five other cases from the archaeological site of Kiskundorozsma–Daruhalom-dűlő II, two cases from Kiskundorozsma–Kettőshatár I, and one case from Kiskundorozsma–Kettőshatár II have been briefly described in a review by Marcsik and her colleagues in 2007 (reference 28 in the revised version of our manuscript). As these cases from the three Kiskundorozsma sites were published in Hungarian, unfortunately, the data are not available for the international readership. (It should also be noted that based on the re-evaluation of the aforementioned cases from the three Kiskundorozsma series, the diagnosis of leprosy is not that definitive in some of them.) Based on the above, we think that every identified prehistoric or historic leprosy case gives us a unique insight into the different manifestations of leprosy and their biological or even social consequences, and can extend our knowledge regarding the origins and geographical distribution of leprosy bacilli, and the migration routes of their human host over time. (As it was demonstrated by Mendum and colleagues in 2018 (reference 25 in the revised version of our manuscript), KD271 already did.) We hope that Reviewer 3 will accept our argumentation.

3) Reviewer 3 commented that “Lines 50 – better to say “infectious disease found mainly in humans”. With that said, I’m curious, is it also transmissible among other primates?”. Following Reviewer 3’s advice, the above-mentioned sentence was rephrased. As for Reviewer 3’s question regarding the transmission of leprosy among other primates, we have to note that unfortunately, we are not specialists in the transmission modes of leprosy among animals as we are palaeopathologists, so we do not know much about how leprosy may be transmitted among other primates. Nevertheless, we searched for information on the internet and found that it may be possible that the disease is transmissible among other primates: “The finding of M. leprae-induced leprosy in wild chimpanzee populations raises the question of the origin(s) of these infections. Mycobacterium leprae is considered a human-adapted pathogen and previous cases of leprosy affecting wildlife were compatible with anthroponosis. Therefore, the prime hypothesis would be human-to-chimpanzee transmission. Potential routes of transmission include direct (such as skin-to-skin) contact and inhalation of respiratory droplets and/or fomites, with the assumption that, in all cases, prolonged and/or repeated exposure is required for transmission11. Chimpanzees at CNP are not habituated to humans and are not approached at distances that would allow for transmission via respiratory droplets. Although these chimpanzees inhabit an agroforest landscape and share access to natural and cultivated resources with humans28, present-day human–chimpanzee direct contact is uncommon. The exact nature of historic human–chimpanzee interactions at CNP remains, however, unknown. For example, robust data on whether chimpanzees were kept as ‘pets’ or were hunted for meat are lacking. Long-term human–chimpanzee coexistence in this shared landscape makes humans the most probable source of chimpanzee infection. However, multiple individuals from several chimpanzee communities across CNP show symptomatic leprosy demonstrating that M. leprae is now probably transmitted between individuals within this population (Hockings et al. 2021).” And “To the best of our knowledge, this is the first paper to report the genomes of nonhuman primate M. leprae strains. While previous studies have shown that M. leprae strains can be transmitted to nonhuman primates, we did not know if naturally occurring leprosy in nonhuman primate was due to incidental infections by human M. leprae strains or by M. leprae strains specific to nonhuman primates. Our results suggest that nonhuman primates, such as chimpanzees and sooty mangabeys in Africa and cynomolgus macaques in Asia, may acquire M. leprae strains from humans as well as transmit these strains between themselves (Honap et al. 2018).” We hope that Reviewer 3 will be satisfied with our answer.

Hockings KJ, Mubemba B, Avanzi C, Pleh K, Düx A, Bersacola E, et al. Leprosy in wild chimpanzees. Nature 2021;598: 652–656. DOI: 10.1038/s41586-021-03968-4 (https://www.nature.com/articles/s41586-021-03968-4#citeas)

Honap TP, Pfister L-A, Housman G, Mills S, Tarara RP, Suzuki K, et al. Mycobacterium leprae genomes from naturally infected nonhuman primates. PLoS Negl Trop Dis 2018;12(1): e0006190. DOI: 10.1371/journal.pntd.0006190 (https://journals.plos.org/plosntds/article?id=10.1371/journal.pntd.0006190)

4) Reviewer 3 advised that “Line 56 – delete “of” before “caused”.” Following Reviewer 3’s suggestion, “of” was deleted in line 56.

5) Reviewer 3 suggested that “Line 72 – replace “hardly” with “minimally”.” Following Reviewer 3’s advice, “hardy” was replaced with “minimally” in line 72.

6) Reviewer 3 commented that “P. 6 – I appreciate the historical background provided, but it would be helpful to provide a few more details about the transmission of leprosy to some other parts of the world to show just how pervasive, transmissible, and fearful it was to humans. A recent paper by Nelson et al. (2022) describe what is purported to be the earliest directly dated case in the Americas. There is also early evidence for the spread of leprosy to India (ca. 4000 BP) (see Robbins et al. 2009, 2013). These are just a few, but there are others that would be worth noting. In addition, given some misunderstandings about leprosy and how it could be contracted, as well as the desire to help those afflicted, there were many leprosaria established in remote or protected areas (individual islands in the Caribbean and Pacific are well-known examples) to sequester and render aid.” Reviewer 3 also provided some references: 1) Nelson, G.C., Dodrill, T.N. and Fitzpatrick, S.M., 2022. A probable case of leprosy from colonial period St. Vincent and the Grenadines, Southeastern Caribbean. International Journal of Paleopathology, 36, pp.7-13.; 2) Robbins, G., Tripathy, V.M., Misra, V.N., Mohanty, R.K., Shinde, V.S., Gray, K.M. and Schug, M.D., 2009. Ancient skeletal evidence for leprosy in India (2000 BC). PloS one, 4(5), p.e5669.; and 3) Robbins Schug, G., Blevins, K.E., Cox, B., Gray, K. and Mushrif-Tripathy, V., 2013. Infection, disease, and biosocial processes at the end of the Indus Civilization. PLoS One, 8(12), p.e84814. We agree with Reviewer 3 that the transmission routes of leprosy to parts of the world is a very interesting topic. However, we think that its detailed discussion would fit better in a review article not our current paper, as it focuses on a particular geographical region (Duna-Tisza Interfluve, Hungary) and time period (Avar Age). As Reviewer 3 mentioned, we already provided the relevant historical background regarding the Avar Age of the Duna-Tisza Interfluve (Hungary) in our manuscript (main text and S1 Text). We are also not sure that it would be suitable to include a detailed discussion on the establishment of leprosaria into our current paper, as in the Avar Age of Hungary, no leprosaria have been established (the first leprosaria of the country were established centuries later). It should also be noted that in another paper from our research group (Spekker O, Tihanyi B, Kis L, Szalontai Cs, Vida T, Pálfi Gy, Marcsik A, Molnár E. 2022. Life and death of a leprosy sufferer from the 8th-century-CE cemetery of Kiskundorozsma–Kettőshatár I (Duna-Tisza Interfluve, Hungary) – Biological and social consequences of having Hansen’s disease in a late Avar Age population from Hungary. PLOS ONE. DOI: 10.1371/journal.pone.0264286), that will be published online in the PLOS ONE in a few days, the case we demonstrated illustrates the social attitude toward leprosy sufferers in early mediaeval Hungary – the social consequences of living and dying with leprosy in the late Avar Age community of Kiskundorozsma–Kettőshatár I, which is also located in the Duna-Tisza Interfluve (Hungary). In this paper, we briefly discuss that our present findings are in accordance with the results of previous studies on other Avar Age cemeteries from the present-day territory of Hungary, indicating that prior to the 13th century CE, lepers have not been segregated from the healthy population, at least in death. To further highlight the social attitude toward leprosy sufferers in early medieval Hungary and to compare it to other time periods (e.g. periods after the 13th century CE, when leprosaria were already established in Hungary), our research group is working on a review article about all the leprosy cases from the Avar Age of Hungary. During the preparation of our aforementioned manuscript, the references Reviewer 3 provided us will be very useful. In summary, we hope that Reviewer 3 will accept our argumentation. We really appreciate the references Reviewer 3 provided us and for sure, will use them in our review article we are working on right now.

7) Reviewer 3 advised that “Line 215 – replace “of” with “in biological anthropology”.” Following Reviewer 3’s suggestion, “of” was replaced with “in” in line 215.

8) Reviewer 3 suggested that “Lines 217-218 – use semicolons to separate sequentially numbered orders”. Following Reviewer 3’s advice, semicolons were used to separate the sequentially numbered orders in lines 217–218.

9) Reviewer 3 noted that “Line 360 – typo - should be “With” (not “Whit”)”. Following Reviewer 3’s comment, the typo was corrected in line 360.

10) Reviewer 3 suggested that “Line 636 – consider revising to: “It is important to note that apart from HD….”.” Following Reviewer 3’s advice, the aforementioned sentence was rephrased in line 636. 

11) Reviewer 3 mentioned that “Line 639 and in several places after (e.g., Lines 670-674). There are a number of cases where longer or multiple sentences are in parentheses. These should usually be reserved for shorter explanations, so consider incorporating these into the text without parentheses.” Following Reviewer 3’s comment, the relevant parentheses were removed and the sentences were incorporated into the text throughout the main document. 

12) Reviewer 3 commented that “Line 674 – replace “decided” with “determined”.” Following Reviewer 3’s suggestion, “decided” was replaced with “determined” in line 674. 

13) Reviewer 3 suggested that “Line 699 – remove double parentheses; instead, parentheses within parentheses should be changed to brackets (e.g., “….arthritis [RA] and psoriatic arthritis [PsA])”. As Reviewer 4 asked us to remove all the abbreviations, there are no double parentheses in the revised version of our manuscript; thus, there is no need to use brackets in the above-mentioned sentence in line 699.

14) Reviewer 3 mentioned that “Line 738 – this is not an adage exclusive of aDNA, but is a commonly used phrase in science and beyond”. Based on Reviewer 3’s comment, the relevant sentence was amended in line 738.

15) Reviewer 3 suggested that “Line 745 – better to say “….in standing and walking and various physical actitivies.”. In the aforementioned sentence, we wanted to highlight that KD520’s difficulties in standing and walking affected her everyday life, i.e., her ability to conduct domestic and occupational physical activities. In our opinion, if we would change the sentence how Reviewer 3 suggested, its meaning would be changed, as well. Therefore, if Reviewer 3 agrees, we would like to keep the sentence in its present form.

16) Reviewer 3 mentioned that “Figure 1 is insufficient for an international audience. It is unclear where this region is located on a regional or global map. Please include one that shows where “A” is found. Map also needs a scale and north arrow.” As it was already indicated in the legend of Figure 1 (“Fig 1. A) Map of Hungary showing the location of the Kiskundorozsma–Daruhalom-dűlő II archaeological site; B) Aerial photo of the Kiskundorozsma–Daruhalom-dűlő II archaeological site; and C) Plan drawing of the Avar Age cemetery of Kiskundorozsma–Daruhalom-dűlő II with the location of the burials of KD271 and KD520. (Contains information from OpenStreetMap and OpenStreetMap Foundation, which is made available under the Open Database License.)”), Figure 1A presents not a region in Hungary but the map of the whole country, and illustrates where the Kiskundorozsma–Daruhalom-dűlő II cemetery, from which the two examined skeletons derive, is located within the country (green dot). In the originally submitted version of our manuscript, there was another map of Hungary and a small map of Europe at the upper left corner in Figure 1A, so the reader could easier figure out the location of Hungary in the continent. However, because of copyright issues, we had to remove this map of Hungary. It was very difficult for us to find another map of Hungary that fits all the copyright requirements of the PLOS ONE – finally, we decided to use the map that was included to the revised version as Figure 1A, as in our opinion, it is suitable to illustrate the location of the Kiskundorozsma–Daruhalom-dűlő II cemetery within Hungary. If Reviewer 3 agrees, we would not like to change the map (Figure 1A) again, as it fits all the requirements of the PLOS ONE regarding maps/satellite images. We know that Hungary is a little country but we hope that most readers of the PLOS ONE know its location in Europe or the globe (and even if not, it can easily and quickly be checked on the internet). Following Reviewer 3’s advice, a scale and an arrow indicating north were put on the map of Hungary (Figure 1A).

17) Reviewer 3 commented that “Figures 2 and others – remove parentheses after letter, which is unnecessary”. Following Reviewer 3’s suggestion, parentheses were removed after the letters in all figures. 

18) Reviewer 4 suggested that “In the line 185 "prevention" should be "salvage.". Following Reviewer 4’s advice, “preventive” was changed to “salvage” in line 185.

19) Reviewer 4 commented that “In lines 275, 288 and 290 the term "turbinate" is used incorrectly. In humans the inferior concha is a separate bone, different from animal "turbinate" bones. A basic anatomical terminology error. Use inferior nasal concha.” Following Reviewer 4’s suggestion, “inferior turbinate bone” was changed to “inferior nasal concha” throughout the main text and relevant supplementary text (S2 Text).

20) Reviewer 4 mentioned that “In the era of electronic (digital, on-line) publications, the use of abbreviations/acronyms saving the cost of printing ink and paper for numerous copies of printed papers saves nothig while it appears to be pretentiously "scientific" and makes reading of manuscripts difficult. I recommend to use full terms instead of pretentious acronymes like TT, LL, HD etc.” Following Reviewer 4’s advice, abbreviations were removed and their fully-written forms were used throughout the main text and supplementary texts.

In the revised version of our manuscript, we tried to execute all suggestions of the reviewers. I hope this new version will be suitable for publication in the PLOS ONE.

Thank you again for the reviewers’ insightful and constructive comments and your editorial work!

Yours sincerely,

Dr Olga Spekker, PhD

Postdoctoral researcher

Department of Biological Anthropology

University of Szeged

Közép fasor 52, H-6726 Szeged, Hungary

Email: olga.spekker@gmail.com

Tel: +36 20 807 72 94

---

## [Editor Report · Decision Letter 3]

2 Mar 2022

The two extremes of Hansen’s disease – Different manifestations of leprosy and their biological consequences in an Avar Age (late 7th century CE) osteoarchaeological series of the Duna-Tisza Interfluve (Kiskundorozsma–Daruhalom-dűlő II, Hungary)

PONE-D-21-21682R3

Dear Dr. Spekker,

We’re pleased to inform you that your manuscript has been judged scientifically suitable for publication and will be formally accepted for publication once it meets all outstanding technical requirements.

Kind regards,

JJ Cray Jr., Ph.D.

Academic Editor

PLOS ONE
---

## [Editor Report · Acceptance letter]

10 Mar 2022

PONE-D-21-21682R3 

The two extremes of Hansen’s disease – Different manifestations of leprosy and their biological consequences in an Avar Age (late 7th century CE) osteoarchaeological series of the Duna-Tisza Interfluve (Kiskundorozsma–Daruhalom-dűlő II, Hungary) 

Dear Dr. Spekker:

I'm pleased to inform you that your manuscript has been deemed suitable for publication in PLOS ONE. Congratulations! Your manuscript is now with our production department. 

Kind regards, 

on behalf of

Dr. JJ Cray Jr. 

Academic Editor

PLOS ONE